# Internet-Delivered Cognitive Behavior Therapy for Young Taiwanese Female Nursing Students with Irritable Bowel Syndrome—A Cluster Randomized Controlled Trial

**DOI:** 10.3390/ijerph16050708

**Published:** 2019-02-27

**Authors:** Tzu-Ying Lee, Tsung-Cheng Hsieh, Huei-Chuan Sung, Wan-Lan Chen

**Affiliations:** 1Institute of Medical Sciences, Tzu Chi University, Hualien 97004, Taiwan; lty@ems.tcust.edu.tw (T.-Y.L.); tchsieh@mail.tcu.edu.tw (T.-C.H.); sung@ems.tcust.edu.tw (H.-C.S); 2Department of Nursing, Tzu Chi University of Science and Technology, Hualien 97004, Taiwan; 3Graduate Institute of Long-term Care & Taiwanese Center for Evidence-based Health Care, Tzu Chi University of Science and Technology, Hualien 97004, Taiwan; 4Department of Human Development and Psychology, Tzu Chi University, Hualien 97004, Taiwan

**Keywords:** Irritable Bowel Syndrome, Internet-delivered cognitive behavioral therapy, expressive writing

## Abstract

Irritable Bowel Syndrome (IBS) is prevalent within the general population. Studies have shown that stress and anxiety co-exist with IBS. Young Taiwanese women commonly exhibit physical and psychological health problems caused by academic stress. The purpose of our current study was to evaluate the efficacy of short-term Internet-delivered cognitive-behavioral therapy (ICBT) on female nursing students in practicum. We performed a cluster randomized controlled trial comprised of 160 participants who met the inclusion criteria, which were divided into three groups: (1) ICBT, (2) expressive writing (EW), and (3) wait-list control. Treatment interventions lasted for 6 weeks. Levels of anxiety, depression, and IBS symptoms were assessed at four time points, baseline assessment at T0, 2 weeks after T0 (T1), at the end of practicum (T2), and at 3-month follow-up (T3). The results showed that ICBT and EW groups exhibited a significant, yet small, reduction in anxiety and depression at T2 and T3 compared to the wait-list control group. The EW group exhibited significantly greater reduction in anxiety and depression compared to the ICBT group at T2. However, the ICBT group demonstrated greater improvements in alleviating anxiety and depression at T3 compared to the EW group. These data indicate that ICBT and EW groups exhibited small effects on anxiety and depression reduction at T2 and T3 compared to the wait-list control group, with no effects on IBS symptoms in young Taiwanese female nursing students.

## 1. Introduction

Irritable Bowel Syndrome (IBS) is one of the most common gastrointestinal disorders with symptoms such as changes in defecation habits, defecation-related abdominal discomfort and pain, bowel dysfunction, and abdominal bloating, which occur in the absence of any structural abnormality [1]. The Rome III criteria for functional GI disorders have served as symptom-based diagnostic criteria for IBS since their release in 2006 and until early 2016. Since then, the Rome Foundation released Rome IV, effectively updating the reference criteria [2]. IBS is a globally prevalent disease ranging from 6–22%, which varies greatly based on research periods and diagnostic criteria. Recent surveys in Taiwan, using Rome I and Rome II criteria, have shown 17.5–22% IBS prevalence. The female to male patient ratio is 2:1, and the age of disease onset is 20–29 years [3]. Many studies indicate higher IBS prevalence among women than men [4]. Women have been reported to experience more psychological distress, and are generally more concerned about their physical conditions than men [5]. In addition, the menstrual cycle indirectly influences defecation, abdominal distention, and abdominal pain [6]. Both physiological and psychological variables play key roles in the etiology of IBS and perpetuate symptoms [7]. Evidence of biological dysregulation in patients with IBS has been reported, and efforts to understand the neurohormonal underpinnings of this disorder are ongoing. However, the exact mechanisms leading to IBS symptoms are not completely understood [8]. Numerous studies have identified psychological aspects, such as personality, stress, and psychological distress as risk factors for IBS onset [9]. One study indicated that while stress is a catalyst that aggravates IBS, anxiety and depression often co-exist with IBS [10]. A systematic review, which compared IBS patients to healthy controls, showed that the former group suffered significantly higher levels of anxiety and depression compared to the latter [11]. Spiller et al. demonstrated that at least half the IBS patients can be described as depressed, anxious, or hypochondriacs. Over 60% of IBS patients receiving tertiary care presented with a psychiatric disorder—most commonly depression, anxiety, or both [12]. Empirical studies have shown that women with IBS exhibit relatively high levels of stress, with stronger emotional reactions, including those related to anxiety and depression [13]. Currently there is minimal research addressing the direct effects of psychological and emotional health on IBS symptom severity.

Although research has shed light on IBS pathophysiology, therapeutic interventions remain symptom driven, employing both pharmacological and non-pharmacological approaches [7]. Several treatment options exist for individuals with IBS, including medication, exercise, fiber supplements, stress management, and psychotherapy [14]. The American College of Gastroenterology (ACG) performed a systematic review to determine the efficacy of 11 IBS therapies—both pharmacological and non-pharmacological—compared to placebo or no treatment. In the ACG highlights, there is evidence supporting the use of antidepressants and psychological therapies for IBS [8]. The British Society of Gastroenterology guidelines on IBS mechanisms and practical management, concluded the need for more accurate means of IBS-constipation / IBS-diarrhea (IBS-C/ IBS-D) identification, which would facilitate appropriate treatment selection [12]. The Cochrane review on the efficacy of psychological interventions found that cognitive behavioral therapy (CBT) and interpersonal psychotherapy may benefit patients with IBS, although some issues such as sample size and clinical heterogeneity required improvement [15]. Another systematic review and meta-analysis, including 30 studies on the effects of psychological therapies on patients with IBS, identified beneficial effects of CBT and other multi-component psychological therapies [16]. Investigating the efficacy of CBT for IBS treatment showed no difference between CBT, relaxation, or standard clinical care [17]. These findings suggest that further investigations are required to better understand the efficacy of psychological therapy for IBS treatment, to better control patient health outcomes.

Several CBT-related strategies are available with respect to treatment type (individual or group therapy). Session duration (30–120 min) varies depending on the therapist, client, and mutual availability. Session quantity (1–12 sessions) and overall duration (ranging from weeks to months) depend on client, therapist, and affordability [18,19]. Over the last 20 years, Internet-delivered CBT (ICBT) has been used to treat over 100 different syndromes and 20 clinical diseases, exhibiting results comparable to those of conventional CBT [20]. The main advantages of ICBT are flexible session times and location, based on Internet access—thereby reducing treatment barriers caused by the lack of availability and distance [20]. Several studies have reported the use of information technology such as: personal digital assistants (PDA), mobile phones, or Internet websites while conducting CBT intervention studies. The results of these studies strongly suggest that Internet use allows effective CBT implementation for IBS treatment [21,22,23].

Previous systematic reviews have indicated that, despite its consistently positive outcomes in reducing IBS symptoms [18], CBT remains inferior when compared to other psychological treatments [19,24]. At present, the long-term maintenance of CBT treatment effects on IBS has not been established [24]. A recent study examining expressive writing (EW) for IBS treatment indicated that EW is useful for reducing anxiety in emotionally expressive patients, while it had the opposite effect for non-emotionally expressive long-term IBS patients [25]. EW efficacy as a treatment method and its ability to address health improvement indicators such as anxiety and related emotional reactions has been confirmed in other health care research [26,27]. 

The purpose of our study was to evaluate ICBT effectiveness for female nursing students during their practicum in Taiwan. This population was selected due to the aforementioned gender differences in incidence and age at IBS onset. Students in the health care profession, such as nursing students, frequently deal with intense academic and practicum stress [28]. To date, only a few studies have investigated the effects of IBS on young women, or evaluated the related intervention methods. One study revealed that 45.9% of students fulfilled IBS diagnosis criteria, while exhibiting high levels of anxiety and depression compared to individuals that do not suffer from IBS [29]. The majority of ICBT programs for IBS are for clinical patients [20], to our knowledge this is the first study to examine ICBT for female students with IBS. We compared the effectiveness of ICBT and EW in the treatment of female nursing practicum students with IBS. The hypothesis is that ICBT exhibits better outcomes compared to other psychological treatments, or wait-list controls, in reducing IBS-related symptoms.

## 2. Materials and Methods

### 2.1. Participants and Eligibility criteria

Study participants were selected from nursing school students using the following inclusion criteria: (1) diagnosis by a general practitioner meeting Rome III criteria [30], which include experiencing repeated abdominal discomfort or pain at least 3 days per month within the preceding 3 months, and experiencing at least two of the following: symptomatic relief after defecation, change in defecation frequency, and/or change in feces consistency; and (2) students about to undergo their first clinical practicum. The exclusion criteria were as follows: (1) students with other types of gastrointestinal diseases, (2) students that had received CBT within the 3 proceeding months, and (3) male students. 

A total of 321 participants (18 to 22 years) completed the IBS screening questionnaire, of which 186 students met the Rome III criteria. A further diagnostic interview by a general practitioner confirmed that 170 participants met the Rome III IBS criteria. As participants were qualified based on the medical interview, they were informed about the study, and their informed consent was requested for participation. Patients were also informed of the voluntary nature of the study—no incentive or extra credit was awarded. Of the 186 interested participants, 161 gave signed consent, while one student chose to withdraw. A total of 160 students participated in the study (Figure 1). The data was collected between July 2014 and July 2015.

### 2.2. Design and Procedure

In Taiwan, a five-year nursing college program requires students to take several short-term practicum courses beginning in the third year towards the end of their fifth year. Following practicum site assignment, nursing students are divided into groups. Due to the college’s program structure, a cluster randomized control trial design was employed. The practicum students in the same practicum unit were randomly assigned to one of the three groups. The number of participants in each group was not equal, as the unit of random assignment to a given intervention or control was by practicum group (cluster). A total of 48 participants were assigned to the ICBT group, which consisted of 13 goal-directed sessions available online for 6 weeks. The EW group consisted of 42 participants, who were required to upload 13 expressive writing essays 100–200 minimum word count within 6 weeks. The wait-list control group consisted of 70 participants who did not receive any ICBT or EW interventions during the research period. This study was conducted according to the provisions of the Declaration of Helsinki (1995), and was approved by Tzu Chi Hospital Ethics Committee (IRB102-110). The research flowchart is illustrated in Figure 1. 

### 2.3. Measures

The Bowel Symptom Severity Scale (BSSS): this scale is primarily used to measure gastrointestinal syndrome frequency, severity level, and influence on lifestyle. Developed by Boyce et al. [31], the BSSS is a 24-item scale measuring eight gastrointestinal related symptoms. The items are scored using a 5-point Likert scale, with scores ranging from 0 (not at all) to 4 (extremely); the maximum score is 96. Cronbach’s α of the scale is 0.88. We obtained consent from the original author and translated the scale into Chinese. The three-procedure approach, suggested by Brislin [32], was adopted for translation. English inventories were first translated by both authors into Chinese, then the Chinese inventory was back-translated into English by a native English speaker. The back-translated version was compared to the original, and the Chinese inventories were then revised as required. Three graduate students were asked to fill out both Chinese and English inventories, and further revisions were made when inconsistent responses were found between the two versions of the same inventory. In the current study, BSSS demonstrated good internal consistency (Cronbach’s α = 0.95 at T0, *N* = 160) and adequate test–retest reliability (*r* = 0.75, *p* < 0.001, assessed at T0 and T1 for the waitlist control group, *N* = 70). 

The State-Trait Anxiety Inventory (STAI-S): the Chinese version of STAI is used primarily to measure state-anxiety levels. In this study, we used the C-STAI adopted by Zhong and Long [33] from the original STAI-S developed by Spielberger et al. [34]. The C-STAI consists of 20 items for scoring, using a 4-point Likert scale, with scores ranging from 1 (not at all) to 4 (extremely). Total scores range from 20–80. Cronbach’s α of the test was 0.86, with test–retest reliability of 0.78 [31].

The Center for Epidemiological Studies Depression Scale (CES-D): the CES-D is primarily used for measuring depression. In this study, we employed the Chinese translation of CES-D [35]. Originally designed in 1997 by Radloff [36], the scale content includes measurements of depression symptoms, including depressive emotions (items 1,3,6,9,10,14,17 and 18), positive emotions (items 4,8,12 and 16), physical symptoms (items 2,5,7,11,13 and 20), and interpersonal problems (items 15 and 19). The scale contains 20 items to be scored using a 4-point Likert scale, ranging from 0 (rarely or none of the time/less than 1 day) to 3 (most of the time/5–7 days). Total scores range from 0–60. Scores in the positive emotion section are reversed. 

Participants who score 16 or higher exhibit depression symptoms. The sensitivity, specificity, and miscalculation values of the Chinese CES-D version were 92.0%, 91.0%, and 8.2%, respectively. The negative predictive value of the scale was 93–100% [35].

### 2.4. Intervention

ICBT group: we developed the ICBT intervention based on Mind Over Mood and made it available to the study participants via the Internet [37] The protocol consists of 13 sessions and encompasses behavioral, emotional, and cognitive components of stress management. The 13 intervention sessions are divided into three parts. Part one: sessions 1–4 focus on behavioral strategies, abdominal breathing, and progressive muscle relaxation training. Part two: sessions 5–7 focus on emotional strategies, performing, and recording pleasant activities. Part three: sessions 8–13 focus on cognitive strategies that teach participants how to recognize negative thoughts and record the recognition process. Intervention sessions were split into two phases, an introductory phase and intervention phase. The introductory phase was conducted 1 week before the participants’ internships. During this phase, individual meetings were arranged to teach participants how to (1) log into the ICBT system, and (2) upload assignments after completing the session. The intervention phase started within the first week of practicum. A total of 13 intervention sessions were performed in this phase. The participants were required to complete and upload their daily assignments before the deadline (before noon each day). Assessments were held at 2 weeks (during intervention), 6 weeks (end of intervention), and 18 weeks after the intervention commenced. 

EW active control group: the EW group protocol was developed based on the Pennebaker and Chung model [26]. Participants were instructed to write about an event during their day when they experienced strong emotions. No additional learning materials or feedback were provided to the participants in this group. The EW assignments were not restricted by format or content but were limited to 200–300 words. A total of 13 EW intervention sessions were scheduled spanning 6 weeks. The EW assignments were uploaded to the system and checked for compliance with the intervention, but no further content evaluation was performed. Assessments were held on the 2nd (T1), 6th (T2), and 18th (T3) weeks of the study, similar to the ICBT group.

Wait list Control group: the control group received no intervention. This group’s progress was recorded on the 2nd, 6th, and 18th weeks. Upon the study’s completion, ICBT was made available to the wait-list control group.

During treatment periods, participants had contact to two therapists: 1) a clinical psychologist with a PhD in clinical psychology, a doctoral student trained in CBT. Contact with therapists was primarily through the school’s e-learning system platform. After completing the assignments, participants were able to send messages to therapists, asking questions regarding specific aspects of the current treatment. Therapists would then reply within one working day. The content uploaded to the Internet platform was stored to ensure that each participant received treatment with similar content.

### 2.5. Analysis

Data were analyzed using descriptive statistics, the generalized estimating equation (GEE), analysis of variance (ANOVA), and analysis of covariance (ANCOVA). The demographic variables, BSSS, STAI-S, and CES-D baselines of the three groups were compared using ANOVA. For each variable, a GEE model including covariates of group, time, group-by-time interaction, and baseline was constructed for evaluating the effects of group, and group-by-time interactions. If the interaction effect occurred, an ANCOVA model including the covariates of group and baseline was conducted at each time point to evaluate group effects. The Bonferroni post-hoc test was then used to compare mean difference between each pair of treatment groups if the group effect reached statistical significance, i.e., *p* < 0.016. Assumptions of data distribution normality and homogeneity of variances were confirmed by the Kolmogorov–Smirnov and Levene tests of equal variances. Statistical significance was set *p* < 0.05 for all comparisons except the Bonferroni post-hoc test. Statistical analyses were performed using SPSS statistics version 20.0 (IBM Corp, Armonk, NY, USA).

## 3. Results

### 3.1. Characteristics of Participants

The average age of the participants was 18.53 years (ICBT: 19.27 years; EW: 19.45 years, and the wait-list control: 18.47 years). The groups were, comprised of fourth year (*n* = 90, 56.3%) and third year (*n* = 70, 43.8%) students in a 5-year junior college program. A total of 160 participants (41.1%) fulfilled the diagnosis criteria for IBS: 67 participants (41.88%) were constipation predominant, 66 (41.25%) were mixed predominant, and 27 (16.87%) were diarrhea predominant. None of the participants reported medication use for their IBS-related problems.

In the BSSS analysis, the ICBT group had a mean score of 24.96 ± 14.27, the EW group had a mean score of 19.07 ± 11.16, and the wait-list control group had a mean score of 32.00 ± 15.94 at baseline (T0). The difference was statistically significant (*p* < 0.001). 

In the STAI-S analysis, the ICBT group had a mean score of 53.00 ± 8.20, the EW group had a mean score of 52.05 ± 9.11, and the wait-list control group had a mean score of 51.89 ± 9.68 at baseline (T0). These three mean scores showed no significant difference (*p* > 0.05). A score of 39–40 was recommended for detecting clinical significance of state anxiety [38]. The mean STAI-S scores indicated that all three groups were in a state of high-anxiety.

In the CES-D analysis, the ICBT, EW, and wait-list control groups had mean scores of 22.42 ± 10.67, 18.81 ± 8.68, and 20.37 ± 10.99, respectively, at baseline (T0). These three mean scores showed no significant difference (*p* > 0.05). The mean scores suggest that all three groups were in mild-depressive states.

Baseline effects were adjusted in the GEE and ANCOVA model for each outcome variables.

### 3.2. Results on Outcome Measures

GEE analysis results (Table 1) revealed an interaction between different groups’ treatment effect and time (group*time) for BSSS (Wald *x*^2^ = 12.22, *p* = 0.002), STATI-S (Wald *x*^2^ = 35.01, *p* < 0.001), and CES-D (Wald *x*^2^ = 35.01, *p* < 0.001). These data indicated the presence of different treatment effects between time points. Therefore, following GEE, the ANCOVA model was conducted to evaluate treatment effects at selected assessment time points (Table 2). 

BSSS assessment of the ICBT group (adjusted mean = −4.66) and EW group (adjusted mean = −4.69) exhibited some improvement compared to the wait-list control group (adjusted mean = − 3.00) on the second week (T1), but it was not statistically significant (ICBT: *β* = −1.66, *p* = 0.341; EW: *β* = −1.69, *p* = 0.373). On the sixth week (T2), the ICBT group (adjusted mean = −2.24) exhibited lack of improvement compared to the wait-list control group (adjusted mean = − 6.72), but the differences were not statistically significant (ICBT: *β* = 4.49, *p* = 0.049; EW: *β* = −1.04, *p* = 0.673). On the 18th week, the ICBT group (adjusted mean = −9.90) and EW group (adjusted mean = −8.22) exhibited considerable improvements compared to the wait-list control group (adjusted mean = −7.04) as assessed using BSSS, while remaining statistically insignificant (ICBT: *β* = −2.87, *p* = 0.136; EW: *β* = −1.18, *p* = 0.569). 

STAI-S assessment of the ICBT group (adjusted mean = −2.26) and EW group (adjusted mean = −5.26) exhibited considerable improvements compared to the wait-list control group (adjusted mean = −1.08) and was statistically significant for the EW group. On the second week (T1), the EW group showed a significantly greater improvement compared to the wait-list control group (ICBT: *β* = −1.17, *p* = 0.477; EW: *β* = −4.18, *p* = 0.016), however only the EW group showed a statistical difference. On the sixth week, the ICBT group (adjusted mean = −4.61) and EW group (adjusted mean = −10.16) exhibited considerable improvements compared to the wait-list control group (adjusted mean = −5.45), the EW group demonstrated a significantly greater change compared to the ICBT and wait-list control groups (ICBT: *β* = 0.84, *p* = 0.638; EW: *β* = −4.71, *p* = 0.012), where only the EW group showed statistical significance. On the 18th week (T3), the ICBT group (adjusted mean = −14.54) and EW group (adjusted mean = −11.96) exhibited considerable improvements compared to the wait-list control group (adjusted mean = -8.40). The ICBT exhibited a greater change compared to the wait-list control group (ICBT: β = −6.15, *p* = 0.002; EW: *β* = −3.56, *p* = 0.089). These data show that reduction in anxiety appeared to take effect at different times in the two intervention groups, while they both exhibited significantly greater improvements compared to the wait-list control group.

CES-D assessment during the second week showed that the ICBT group (adjusted mean = −2.28) and EW group (adjusted mean = −2.12) exhibited considerable improvements compared to the wait-list control group (adjusted mean = 1.68). The ICBT group changed significantly more compared to the wait-list control group (ICBT: *β* = −3.96, *p* = 0.011; EW: *β* = −3.80, *p* = 0.019). On the sixth week, the ICBT group (adjusted mean = −3.28) and EW group (adjusted mean = −6.18) exhibited considerable improvements compared to the wait-list control group (adjusted mean = −1.40). Furthermore, the EW group showed significantly greater change compared to the wait-list control group (ICBT: *β* = −1.87, *p* = 0.254; EW: *β* = −4.77, *p* = 0.006). At T3 the ICBT group (adjusted mean = −9.15) and EW group (adjusted mean = −6.88) exhibited reduction in depression compared to the wait-list control group (adjusted mean = −3.44). Effects of intervention in the ICBT group were most apparent at T3, and were statistically significant (ICBT: *β* = −5.71, *p* < 0.001; EW: *β* = −3.44, *p* = 0.041). These findings suggest that with respect to depression, intervention effects depend on treatment duration or the number of intervention sessions (Figure 2 and Figure 3).

## 4. Discussion

The cluster randomized controlled trial was conducted to compare the effects of Internet delivered cognitive therapy (ICBT), with self-administered expressive writing (EW), and wait-list control on IBS and its related emotional symptoms. The results indicated that IBS symptoms, depression, and anxiety decreased over time for all three tested groups with one exception, the ICBT group’s IBS scores increased slightly at the end of treatment. We speculated that ICBT’s self-help learning may create additional stress when students have heavy assignment loads that must be completed shortly after the end of practicum. This may explain why IBS scores increased for the ICBT group at T2. Contrary to our hypothesis, no significant differences were observed between the three groups, indicating that neither ICBT nor EW was superior to the wait-list control in easing IBS symptoms. A similar conclusion was reported by a recent meta-analysis [39], Internet-delivered CBT, or minimal contact CBT, showed no improvement in IBS symptoms. The authors speculated that periodically direct contact between therapists and patients/clients, in addition to the therapists assistance and reflection were necessary for effective psychological therapy. A number of controlled trial studies have demonstrated that CBT—among other personal contact psychological therapies—is beneficial for alleviating IBS symptoms [40,41]. Another explaination is the length of interevention time, and the number of psychotherapy sessions attended by the IBS sufferer. CBT requires consistent guided effort and focus, of which 8–10 weeks is insufficient for accurate assessment of its possible effects on IBS.

ICBT may effectively transfer the mechanism of control to the IBS sufferer, which allows them to manage their anxiety and depression, unfortunately, without affecting their IBS symptoms. The results of this study are similar to the Boyce et al. study [17] which supports that the proposed cognitive model appears to have significant effects in reducing emotions of anxiety and depression, though there were no significant differences among treatments in their results. The Cochrane review of non-pharmacological interventions for functional syndromes (14 of the 21 RCTs used CBT) showed small positive effects when CBT was learnt by participants [42]. It is known, that at least half of IBS patients suffer from anxiety, depression or hypochondrisis. Tertiary care studies suggested that patients with IBS and other funcitonal bowel disorders were more likely to be diagnosed with psychiatirc disorders,-most commonly anxiety or depressive disorder [43,44,45]. The cognitive mechanism of effect between CBT and the emotional symptoms of anxiety and depression has been widely studied and supported. Based on recent research [46] exploring neuroplasticity and the effect of a trained mind on the physical, mental and emotional experience of an individual remains paramount, suggests CBT, when consistently practiced, may prove to be a powerful tool in the health care arsenal for both physican and patient.

We observed greater differences between ICBT and the other two methods, particularly on week 18, where ICBT exhibited notable efficacy regarding anxiety and depression. Our study results are in accordance with previous published data [47,48]. Findings from a recent meta-analysis examining the neural basis of CBT, and its effects on symptoms of anxiety and depression, showed that CBT exerts its effects on anxiety and depression by enhancing the top–down cognitive control of emotions (e.g., prefrontal regions) and/or decreasing the bottom–up flow of fear-related emotions from limbic structures [49]. Cognitive–behavioral control may lead to long term increased awareness, improved emotional health, and con-concomitant physical health [50]. 

EW appears to be significantly effective at short-term reduction of both anxiety and depression. On the second and sixth weeks, the EW group exhibited greater anxiety relief compared to the ICBT and control groups. However, the ICBT group showed some depression relief within the first two weeks compared to the waitlist control. In addition, on the sixth week, the EW group exhibited greater anxiety improvement than the ICBT group. These results indicate that EW can effectively lower anxiety, which concurs with research published by Niles et al. [51]. This could be attributed to EW enabling young women to relieve stress and anxiety-related emotions with significant efficacy on a short-term basis. The maintenance of this form of intervention effect could not be confirmed after the study was concluded. Although ICBT effects were not apparent for anxiety and depression in the short-term, it superseded EW by the end of the intervention interval. 

Interestingly, the EW and wait-list control groups exhibited a similar decline over the research period, which may be due to IBS characteristics. Students with IBS develop physiological and psychological symptoms in environments with high practicum stress. However, post-practicum, when stress is relieved, physiological and psychological symptoms also decrease. ICBT teaches behavioral, emotional, and cognitive control, therefore, observing its potential efficacy in the short-term is impractical. Learning and implementing behavioral strategies to lower anxiety and depression requires time, consistent practice, and guided professional therapy. Individuals implementing the skills necessary for reducing anxiety, depression, and other related emotions, attain observable and measurable effects over time. 

### Study Limitations

The scope of our research was limited by several factors: our samples consist of a homogeneous population with respect to demographic characteristics and educational background, which may limit the generalizability of the research findings. Therefore the study should be repeated using more diverse clinical or community samples. It is worth noting that the inclusion criteria used in this study was Rome III, however, the Rome IV criteria for IBS diagnosis are currently in use. The cluster randomized control trial used in the study—though overcomes the possible contamination effects, which are likely to occur by contact among individual participants in different groups—can be susceptible to some methodological problems, such as the imbalance in the number of participants between wait-list control and treatment groups. It is unclear whether participants in this study meet the Rome IV criteria for IBS. Inadequate participant blinding is another limitation, which may result in placebo or nocebo effects on treatment outcomes. Furthermore, data on IBS, depression, and anxiety were derived from self-report questionnaires. The use of multimethod examination for symptom assessment would provide a base for stronger conclusions.

## 5. Conclusion

ICBT has a positive effect on reducing anxiety and depression in young Taiwanese female nursing students, while exhibiting no effect on IBS symptoms. EW and ICBT show similar effects short-term, but ICBT performs better over the longer term.

## Figures and Tables

**Figure 1 ijerph-16-00708-f001:**
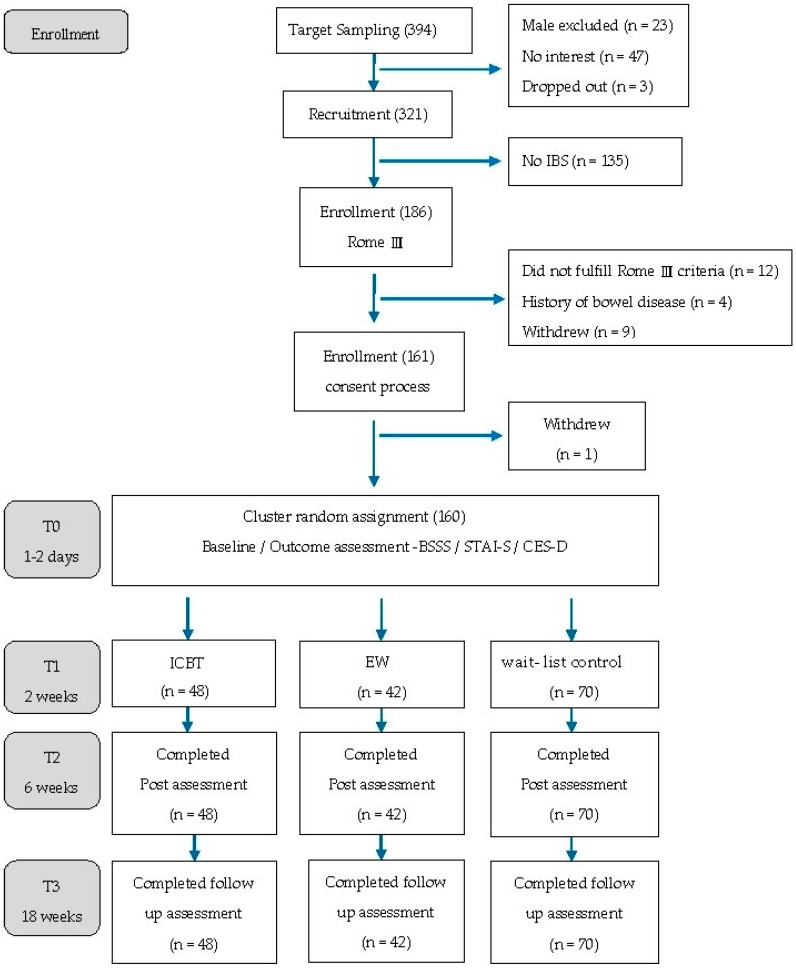
Study Flowchart.

**Figure 2 ijerph-16-00708-f002:**
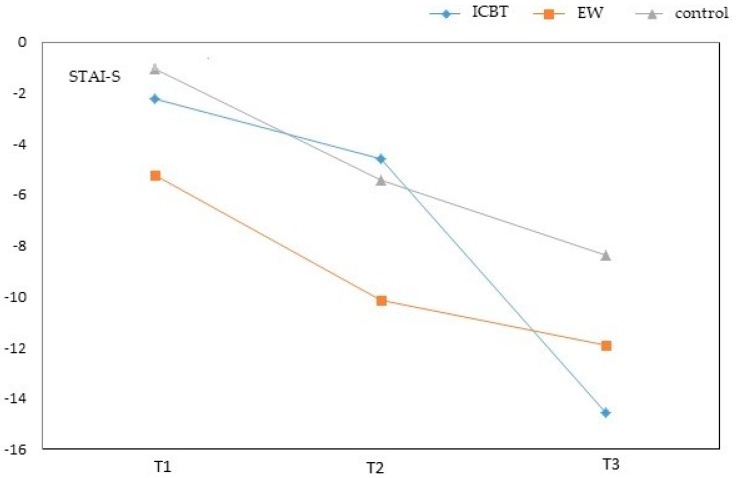
Adjustment mean change of STAI-S over time. Adjustment mean was estimated using the ANCOVA model in Table 2. T1 = assessment at 2 weeks, T2 = assessment at 6 weeks and end of practicum, and T3 = assessment at 18 weeks.

**Figure 3 ijerph-16-00708-f003:**
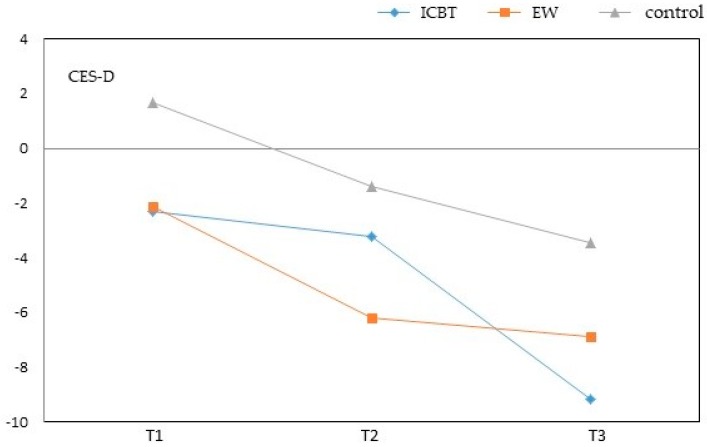
Adjustment mean change of CES-D over time. Adjustment mean was estimated using the ANCOVA model in Table 2. T1 = assessment at 2 weeks, T2 = assessment at 6 weeks and end of practicum, and T3 = assessment at 18 weeks.

**Table 1 ijerph-16-00708-t001:** GEE model results of IBS symptoms (BSSS), anxiety (STAI-S) and depression (CES-D) by group and time interaction (*N* = 160).

	IBS Symptoms (BSSS)			Anxiety (STAI-S)			Depression (CES-D)		
Effect	wald *x*^2^	df	*P*	wald *x*^2^	df	*P*	wald *x*^2^	df	*P*
Intercept	41.68	1	<0.001	27.53	1	<0.001	18.49	1	<0.001
Group	7.20	2	0.027	267.20	2	<0.001	13.96	2	<0.001
Time	54.06	1	<0.001	112.04	1	<0.001	79.98	1	<0.001
Group*Time	12.22	2	0.002	35.01	2	<0.001	194.87	2	<0.001
Baseline	122.83	1	<0.001	34.73	1	<0.001	23.70	1	<0.001

BSSS = Bowel Symptom Severity Scale, STAI-S = State-Trait Anxiety Inventory, CES-D = Center for Epidemiological Studies Depression Scale, Intercept = intercept term, Group = treatment group, Time = time points, Group*Time = interaction term between treatment group and time points, Baseline = BSSS, STAI-S, and CES-D measures at baseline (T0), wald *x*^2^ = test of hypotheses on parameters estimated by maximum likelihood, df = degrees of freedom.

**Table 2 ijerph-16-00708-t002:** ANCOVA results for change in IBS symptoms (BSSS), anxiety (STAI-S), and depression (CES-D) at each time.

	T1					T2					T3				
Variables	Adjusted mean	β	SE	*P*	Post hoc	Adjusted mean	β	SE	*P*	Post hoc	Adjusted mean	β	SE	*P*	Post hoc
BSSS															
ICBT	−4.66	−1.66	1.74	0.341		−2.24	4.49	2.26	0.049		−9.90	−2.87	1.91	0.136	
EW	−4.69	−1.69	1.89	0.373		−7.76	−1.04	2.46	0.673		−8.22	−1.18	2.08	0.569	
Control	−3.00					−6.72					−7.04				
Baseline		−0.35	0.05	<0.001			−0.427	0.065	<0.001			−0.441	0.055	<0.001	
STAI-S															
ICBT	−2.26	−1.17	1.65	0.477		−4.61	0.84	1.78	0.638		−14.54	−6.15	2.00	0.002	1 > 3 *
EW	−5.26	−4.18	1.71	0.016	2 > 3 *	−10.16	−4.71	1.85	0.012	2 > 1 * 2 > 3 *	−11.96	−3.56	2.08	0.089	
Control	−1.08					−5.45					−8.40				
Baseline(ref.)		−0.331	0.077	<0.001			−0.502	0.083	<0.001			−0.677	0.093	<0.001	
CES-D															
ICBT	−2.28	−3.96	1.54	0.011	1 > 3*	−3.28	−1.87	1.64	0.254		−9.15	−5.71	1.60	<0.001	1 > 3 **
EW	−2.12	−3.80	1.60	0.019		−6.18	−4.77	1.70	0.006	2 > 3 *	−6.88	−3.44	1.67	0.041	
Control	1.68					−1.40					−3.44				
Baseline(ref.)		−0.323	0.063	<0.001			−0.438	0.067	<0.001			−0.621	0.065	<0.001	

BSSS = Bowel Symptom Severity Scale, STAI-S = State-Trait Anxiety Inventory, CES-D = Center for Epidemiological Studies Depression Scale, ICBT = Internet-delivered cognitive behavioral therapy group, EW = expressive writing group, control = wait-list control group, baseline = T0, T0 = baseline assessment, T1 = assessment at 2 weeks, T2 = assessment at 6 weeks and end of practicum, T3 = assessment at 18 weeks, * *p* < 0.05, ** *p* < 0.01.

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
