# Peer review of "Internet-Delivered Cognitive Behavior Therapy for Young Taiwanese Female Nursing Students with Irritable Bowel Syndrome—A Cluster Randomized Controlled Trial"

_ijerph, 2019, doi:10.3390/ijerph16050708_

Reviewer 1 Report

Dear Dr. Ristić,

 Thank you for asking me to Review this study again. 
 The article has improved but there are still a number of important issues with the article and its discussion /conclusion as discussed below 

line 2 
 Your study is not about young women so please change the title to something like 
Internet-delivered cognitive behavior therapy for young Taiwanese female nursing students with Irritable Bowel Syndrome. A cluster randomized controlled trial 

 22 
exhibited significantly some reduction 
 Do you mean exhibited a significant but small reduction 

 Line 26 
 You cannot state that ICBT can effectively reduce anxiety and depression if in line 22 you stated that The ICBT and EW groups exhibited significantly some reduction in anxiety and depression at T2 and T3 compared to the wait-list control group.

 Line 27
with no effect on IBS symptoms in young women. 
 You didn't study young women so please change young women to 
young Taiwanese female nursing students. 

 line 228 
The second sentence is too long and grammatically incorrect please change:
The average age of the participants was 18.53 ± 1.11 years. The ICBT group was 19.27 ± 0.45 years, the EW group was 19.45 ± 0.55 years, and the wait-list control group was 18.47 ± 0.70 years. to:
The average age of the participants was 18.53 years (ICBT: 19.27 years; EW: 19.45 years, and the wait-list control group: 18.47 years).

 237 
. These three mean scores differed significantly (F=10.90, p<0.001). 
 you probably mean the difference was statistically significant (p<0.001).< span="">

 240 
The mean scores indicated that all three groups were in a state of high-anxiety. please give a reference and explain what the reference scores are for normal, mild, moderate and high-anxiety.

 256 
but was not statistically significant please change to
but this was not statistically significant

 266 
and was statistically significant please change to
and this was statistically significant

 316 
with only one exception, the IBS scores increased slightly at the end of the treatment for the ICBT group. 
 why do you ignore that at all your measurement points there was no statistically significant difference between the three groups that also means as I wrote in my previous Review that the scores for ICBT at T2 and T3 in your figure 2 are misleading because there was no difference with the two other group 

318
 self-help learning of ICBT may become additional stress while students usually had a great deal of assignments need to be completed shortly after the end of practicum
please change to
self-help learning of ICBT may create additional stress while students usually had a great deal of assignments that needed to be completed shortly after the end of practicum
 also I can't follow this logic because if you follow it through than the same should have happened in the EW group and that wasn't the case 

323
in which CBT delivered through the internet, or minimal contact CBT showed  no improvement in IBS symptoms.The authors speculated that direct contact between therapists and patients/clients was necessary for psychological therapy to be effective. 
 this is a strange conclusion because in minimal contact CBT there is direct contact between therapists and patients/clients

 328 
 you conclude that 8 to 10 weeks is insufficient. However Van Dulmen et al. (7) compared group cognitive therapy with a wait-list controlled group. The group therapy consisted of eight sessions of cognitive behavior therapy in a group setting. Although there were limitations of the study they concluded that CBT is effective for IBS 

 332 
The results of this study are similar to the Boyce study [17] which supports that the proposed cognitive model appears to have its effect by altering the cognitive response to visceral hypersensitivity. 
The second part of this sentence doesn't make any sense as both your study and Boyce have shown that CBT is not effective for IBS. In other words it cannot have its effect by altering the cognitive response to visceral hypersensitivity because it is not effective. 

333
The Cochrane review of non-pharmacological interventions for functional syndromes (14 of the 21 RCTs used CBT) showed excellent effect when CBT was learned by the participants [41]. 
There are a number of issues with this sentence.
 your reference 41 is the protocol not the article 
 the Cochrane review did not show excellent effects of CBT; it showed that CBT had a small effect on somatic symptoms compared with usual care or waiting-list conditions but CBT was not more effective if an active control group was used. 

 336 
Hypochondriasis is a mental disorder so what do you mean by 
or some form of hypochondriacal manifestation of mental illness.

 337 
Studies from terriary care suggested that more than 60% of IBS sufferers were in fact suffering from other psychiatric disorders- 
 do you mean comorbid psychiatric disorders because IBS is not a psychiatric disorder ? Also it is tertiary care not terriary care 

339 
The cognitive mechanism of effect between CBT and the emotional symptoms of anxiety and depression are currently being researched.
 what do you mean by that? that we do not know how CBT can alleviate anxiety and depression and we are trying to find that out? 

 340 
Understanding the underpinnings of the possibility that the mind affects the physical experience and health within an individual suggests strongly, that CBT, when it is also similarly refined in its practice, may prove to be a powerful tool in the health care arsenal for both physican and patient. 
 understanding the underpinnings of the possibility does not suggest anything let alone strongly. 

 344 
Greater differences between ICBT and the other two methods were observable particularly at the 18th week, where ICBT exhibited notable efficacy. 
 this sentence is misleading it suggest that ICBT was more effective at all stages which is not the case; also I presume you talk about anxiety and depression but please mention it then 

 345 
The results of this study are in agreement with the results of previous studies [45, 46, 47, 48]. ICBT appears to exert its effects on anxiety and depression by enhancing the top-down cognitive control of emotions (e.g. prefrontal regions) and/or decreasing the bottom flow of fear-related emotions from limbic structures. Cognitive-behavioral control appears to lead to increased awareness, improved emotional health and con-comittant physical health over the long term. 

ICBT appears to exert its effects where does that come from because you can't conclude that from your own study. and the same applies to Cognitive-behavioral control appears to

 352 
At the second and sixth weeks, the EW group exhibited greater anxiety relief than did the ICBT and control groups. However, greater relief from depression was found within the first 2 weeks for the ICBT. In addition, in the sixth week, the EW group exhibited greater IBS improvement than the ICBT group. 

 the difference in depression relief within the first two week was minimal for ICBT compared to EW 
 at all three points there were no statistically significant differences between the groups on IBS symptoms therefore you cannot state that EW exhibited greater IBS improvement than the ICBT group. 

 353 
However, greater relief from depression was found within the first 2  weeks for the ICBT. 
 At baseline the depression score in the ICBT group was 22.42 
 At T1 the ICBT group had improved by 0.16 more than the EW group; this minute difference doesn't warrant your conclusion of greater relief 

 357 
The  maintenance of this form of intervention effect could not be established after the research had ended.
 what do you mean by could not be established? do you mean that the intervention effect disappeared after the intervention ended? or do you mean that you did not check or could not check (what was the reason?) after the intervention ended ? 

 364 
ICBT teaches behavioral, emotional, and cognitive control, however observing its potential efficacy in the short term is impractical.
 your results showed that in the beginning ICBT had a negative effect on anxiety and depression. there was nothing impractical about observing etcetera it was simply not effective 

 367 
Over the long term, individuals implementing the skills necessary to reduce their anxiety, depression,
 ICBT was starting to take effect between T2 and T3; or in other words from six weeks to 18 weeks; one of the conclusions of one of your references (the Cochrane review of therapy for MUS) was that trials should conduct      follow-up assessments   until   at      least   one     year    after   the     end     of      treatment.       That would be longer term follow-up; 18 weeks is not long-term it is simply 12 weeks after the end of treatment 

371 
This research was limited in its scope due to several factors: Our sample consists of a relatively homogeneous population with respect to demographic characteristics and educational background, and it may limit the generalizability of the research findings.
 it wasn't a relatively homogeneous population it was a very homogeneous population as it only included young female nurse students. 

373
 The exclusion criteria ( such as other type of gastrointestinal diseases diagnosis or receiving CBT within 3 months) used in this study also raised some concerns regarding results generalizability. these two exclusion criteria do not raise concerns about generalisability; it is common sense to exclude those two.

 377 
The randomized control trials we used in the study, though overcome the possible contamination effects, which is likely to occur by contact among individual participants in different groups, it can be susceptible to some methodological problems, such as the number of participants imbalance between treatment groups.
 first of all this sentence is way too long and secondly it doesn't make any sense. randomisation does not overcome the possible contamination effects even more so because it was an unblinded trial using subjective outcomes.

 382
 Due to the length of the study,
 you need to change the start of the sentence because this way it doesn't make any sense 
 why would it be more useful to increase frequency of interventions and extend the follow-up? this is the section about study limitations; so why is the frequency of interventions and the length of the study a limitation? 
 why don't you mention that a no treatment control group is a limitation and the same applies to subjective outcomes in an unblinded trial 
 another limitation is that the patients in the no treatment group had much higher IBS scores and therefore were much more ill 

 a major limitation of the study was the use of a waitlist control group and the fact that the groups were poorly matched as shown by the unevenly distribution of participants (48, 42 and 70)
 also that (227/394) participants were excluded from the study 

386 
ICBT can effectively reduce anxiety and depression in young women
 you can't actually say that based on your study because your study was not done on young women 
 please change that to 
ICBT can reduce symptoms of anxiety and depression in young Taiwanese female nursing students with mild symptoms 

Author Response

Thank you for your very careful review of our manuscripts, and for the comments, corrections and suggestions that ensued. Your comments helped us refine our paper by clarifying critical sections and presenting results. Please refer to the manuscript for numerous changes motivated by your comments. Below we summarize major comments you provided and our reaction to them.

Point 1: Your study is not about young women so please change the title to something like Internet-delivered cognitive behavior therapy for young Taiwanese female nursing students with Irritable Bowel Syndrome. A cluster randomized controlled trial

Response 1: Thanks for the suggestion. The title of this paper has been changed to  “Internet-delivered cognitive behavior therapy for young Taiwanese female nursing students with Irritable Bowel Syndrome - A cluster randomized controlled trial

Point 2: line 22 exhibited significantly some reduction.  Do you mean exhibited a significant but small reduction 

Response 2: Thanks for the suggestion. In the revision, the sentence has been revised as suggested.

The ICBT and EW groups exhibited a significant but small reduction in anxiety and depression at T2 and T3 compared to the wait-list control group. (line 23-25)

Point 3:  Line 26  You cannot state that ICBT can effectively reduce anxiety and depression if in line 22 you stated that The ICBT and EW groups exhibited significantly some reduction in anxiety and depression at T2 and T3 compared to the wait-list control group.

Response 3: Thanks for pointing this out. In the revision, the sentence has been revised as suggested.

The ICBT and EW groups demonstrated small effects in reducing anxiety and depression at T2 and T3 compared to the wait-list control group, (line 27-29)

Point 4:  Line 27 with no effect on IBS symptoms in young women.  You didn't study young women so please change young women to young Taiwanese female nursing students. 

Response 4: Thanks for the suggestion. In the revision, the sentence has been revised as suggested.

…in young Taiwanese female nursing students. (line 29-30)

Point 5:  line 228 The second sentence is too long and grammatically incorrect please change: The average age of the participants was 18.53 ± 1.11 years. The ICBT group was 19.27 ± 0.45 years, the EW group was 19.45 ± 0.55 years, and the wait-list control group was 18.47 ± 0.70 years. to:
The average age of the participants was 18.53 years (ICBT: 19.27 years; EW: 19.45 years, and the wait-list control group: 18.47 years).

Response 5: Thanks for the suggestion. In the revision, the sentence has been revised as suggested.

The average age of the participants was 18.53 years (ICBT: 19.27 years; EW: 19.45 years, and the wait-list control: 18.47 years). (line 231-232)

Point 6: line 237 These three mean scores differed significantly (F=10.90, p<0.001).  you probably mean the difference was statistically significant (p<0.001).

Response 6 : Thanks for the suggestion. In the revision, the sentence has been revised as suggested.

The difference was statistically significant (p<0.001). (line 239)

Point 7: line 240 The mean scores indicated that all three groups were in a state of high-anxiety. please give a reference and explain what the reference scores are for normal, mild, moderate and high-anxiety.

Response 7: Thanks for the suggestion. We have provided a reference for interpreting the scores of STAI-S. Based on the cut point of STAI-S, the three groups were in high state anxiety.   

A score of 39-40 was recommended for detecting clinical significance of state anxiety . (Line 242-243)

Point 8: line 256 but was not statistically significant please change to but this was not statistically significant

Response8: Thanks for pointing this out. In the revision, the sentence has been revised as suggested.

…but this was not statistically significant (Line 258)

Point 9: line 266 and was statistically significant please change to and this was statistically significant.

Response 9: Thanks for pointing this out. In the revision, the sentence has been revised as suggested.

…and this was statistically significant (Line 267)

Point 10: line 316  with only one exception, the IBS scores increased slightly at the end of the treatment for the ICBT group. why do you ignore that at all your measurement points there was no statistically significant difference between the three groups that also means as I wrote in my previous Review that the scores for ICBT at T2 and T3 in your figure 2 are misleading because there was no difference with the two other group

Response 10: Thanks for the comments. Indeed, the figure 2 might be misleading. We removed the figure in the revision.

As we discussed in the reply for the first revision, We did highlight the effect of ICBT was not significant on IBS. As it can be seen at line 322-324, “Contrary to our hypothesis, no significant differences were observed between the three groups, indicating that neither ICBT nor EW was superior to the wait-list control to ease IBS symptoms.”

Point 11: line 318 self-help learning of ICBT may become additional stress while students usually had a great deal of assignments need to be completed shortly after the end of practicum.please change to
self-help learning of ICBT may create additional stress while students usually had a great deal of assignments that needed to be completed shortly after the end of practicum .

Response 11: Thanks for the suggestion. Modified as suggested. (Line 320)

Point: 12 also I can’t I can't follow this logic because if you follow it through then the same should have happened in the EW group and that wasn't the case

Response 12: Thanks for the thoughtful comments. Unlike the ICBT group which received both learning materials and feedback online, the EW group did not receive any further validation or feedback on their emotional writing. We have added this description of EW in the section of intervention.

No additional learning materials or feedback were provided to the participants in this group. (Line 199-200)

Point 13: line 323 in which CBT delivered through the internet, or minimal contact CBT showed no improvement in IBS symptoms.The authors speculated that direct contact between therapists and patients/clients was necessary for psychological therapy to be effective. this is a strange conclusion because in minimal contact CBT there is direct contact between therapists and patients/clients

Response 13: Thanks for pointing this out. We have revised this sentence and made the content clearer.

The authors speculated that periodically direct contact between therapists and patients/clients and the assistance and reflection of the therapist were necessary for psychological therapy to be effective. (Line 326-328)

Point 14: line 328  you conclude that 8 to 10 weeks is insufficient. However Van Dulmen et al. (7) compared group cognitive therapy with a wait-list controlled group. The group therapy consisted of eight sessions of cognitive behavior therapy in a group setting. Although there were limitations of the study they concluded that CBT is effective for IBS.

Response 14: Thank you for bringing that to our attention. Our research participants were only females with mean age of 18-19. Comparison to the Dulmen et al research, with a median age of 40 does not allow for an accurate association of the potential efficacy of CBT. Should younger people affected with IBS and consistently implement CBT to reduce their concomitant emotional and psychological suffering due to IBS symptoms, this would make for an interesting longitudinal cohort study.

Point 15: line 332 The results of this study are similar to the Boyce study [17] which supports that the proposed cognitive model appears to have its effect by altering the cognitive response to visceral hypersensitivity. The second part of this sentence doesn't make any sense as both your study and Boyce have shown that CBT is not effective for IBS. In other words it cannot have its effect by altering the cognitive response to visceral hypersensitivity because it is not effective. 

Response 15: Thank you for pointing this point. In the revision, we have revised the sentences and made the content clearer.

ICBT may effectively transfers the mechanism of control to the IBS sufferer, which allows them to manage their anxiety and depression, unfortunately, without affecting their IBS symptoms. The results of this study are similar to the Boyce, et al study [17] which supports that the proposed cognitive model appears to have significant effects in reducing emotions of anxiety and depression, though there were no significant differences among treatments in their results. (Line 333-337)

Point 16: line 333 The Cochrane review of non-pharmacological interventions for functional syndromes (14 of the 21 RCTs used CBT) showed excellent effect when CBT was learned by the participants [41]. There are a number of issues with this sentence. your reference 41 is the protocol not the article the Cochrane review did not show excellent effects of CBT; it showed that CBT had a small effect on somatic symptoms compared with usual care or waiting-list conditions but CBT was not more effective if an active control group was used. 

Response 16: Thanks for pointing this out. In the revision, the content and reference were modified.   

The Cochrane review of non-pharmacological interventions for functional syndromes (14 of the 21 RCTs used CBT) showed small positive effect when CBT was learned by the participants [42]. (Line 337-339)

Reference:

42. Van Dessel N, Den Boeft M, van der Wouden JC, Kleinstäuber M, Leone SS, Terluin B,Numans ME, van der Horst HE, van Marwijk H. Non-pharmacological interventions for somatoform disorders and medically-unexplained physical symptoms (MUPS) in adults. Cochrane Database Syst Rev. 2014, Nov. 1, 11, CD011142. DOI: 10.1002/14651858.CD011142.pub2.

Point 17: line 336 Hypochondriasis is a mental disorder so what do you mean by or some form of hypochondriacal manifestation of mental illness.

Response 17: Thanks for pointing this out. We have change the hypochondriacal manifestation of mental illness to hypochondriasis. (Line 340)   

Point 18: line 337 Studies from terriary care suggested that more than 60% of IBS sufferers were in fact suffering from other psychiatric disorders-  do you mean comorbid psychiatric disorders because IBS is not a psychiatric disorder ? Also it is tertiary care not terriary care 

Response 18: Thanks for pointing this out. We have revised the sentences, simplified the language, and made the content clearer.

Tertiary care suggested that patients with IBS and other funcitonal bowel disorder were more likely to be diagnosed with psychiatirc disorder-most commonly anxiety or depressive disorder. (Line 340-342)

Point 19: line 339 The cognitive mechanism of effect between CBT and the emotional symptoms of anxiety and depression are currently being researched.  what do you mean by that? that we do not know how CBT can alleviate anxiety and depression and we are trying to find that out? 

Response 19: Thanks for pointing this out. We have revised the sentences and made the content clearer.

The cognitive mechanism of effect between CBT and the emotional symptoms of anxiety and depression has been widely studied and supported. (Line 342-344)

Point 20: line 340 Understanding the underpinnings of the possibility that the mind affects the physical experience and health within an individual suggests strongly, that CBT, when it is also similarly refined in its practice, may prove to be a powerful tool in the health care arsenal for both physician and patient.  understanding the underpinnings of the possibility does not suggest anything let alone strongly. 

Response 20: Thanks for the comment. We have revised the relevant section and made the content clearer. 

Based on recent research [46] exploring neuroplasticity and the effect of a trained mind on the physical, mental and emotional experience of an individual remains paramount, suggests CBT, when consistently practiced, may prove to be a powerful tool in the health care arsenal for both physican and patient. (Line 344-347)

 Point 21: line 344 Greater differences between ICBT and the other two methods were observable particularly at the 18th week, where ICBT exhibited notable efficacy. this sentence is misleading it suggest that ICBT was more effective at all stages which is not the case; also I presume you talk about anxiety and depression but please mention it then. 

Response 21: Thanks for pointing this out. Indeed, the sentence can be misleading. It has been revised as suggested.

where ICBT exhibited notable efficacy on anxiety and depression. (line 349)

Point 22: line 345 The results of this study are in agreement with the results of previous studies [45, 46, 47, 48]. ICBT appears to exert its effects on anxiety and depression by enhancing the top-down cognitive control of emotions (e.g. prefrontal regions) and/or decreasing the bottom flow of fear-related emotions from limbic structures. Cognitive-behavioral control appears to lead to increased awareness, improved emotional health and con-comittant physical health over the long term. 

ICBT appears to exert its effects where does that come from because you can't conclude that from your own study. and the same applies to Cognitive-behavioral control appears to

Response 22:Thanks for the comments. We have revised the relevant section and made the content clearer. 

 The results of this study are in agreement with the results of previous studies [47, 48]. Findings from a recent meta-analysis examining the neural basis of CBT on symptoms of anxiety and depression and showed that CBT appears to exert its effects on anxiety and depression by enhancing the top-down cognitive control of emotions (e.g. prefrontal regions) and/or decreasing the bottom-up flow of fear-related emotions from limbic structures [49]. Cognitive-behavioral control may lead to increased awareness, improved emotional health and con-concomitant physical health over the long term [50]. (Line 349-355)

Point 23: line 352 At the second and sixth weeks, the EW group exhibited greater anxiety relief than did the ICBT and control groups. However, greater relief from depression was found within the first 2 weeks for the ICBT. In addition, in the sixth week, the EW group exhibited greater IBS improvement than the ICBT group. 

 the difference in depression relief within the first two week was minimal for ICBT compared to EW at all three points there were no statistically significant differences between the groups on IBS symptoms therefore you cannot state that EW exhibited greater IBS improvement than the ICBT group. 

Response 23Thanks for pointing this out. Indeed, EW group exhibited greater anxiety improvement than the ICBT group at the sixth week, not IBS. It has been revised.

...in the sixth week, the EW group exhibited greater anxiety improvement than the ICBT group. (Line 359-360)

Point 24 line 353 However, greater relief from depression was found within the first 2 weeks for the ICBT. At baseline the depression score in the ICBT group was 22.42 At T1 the ICBT group had improved by 0.16 more than the EW group; this minute difference doesn't warrant your conclusion of greater relief 

Response 24: Thanks for pointing this out. We have revised the sentences and made the content clearer.

However, some relief from depression was found within the first 2 weeks for the ICBT group compared to the waitlist control. (Line 358-359)

Point 25: line 357 The maintenance of this form of intervention effect could not be established after the research had ended. what do you mean by could not be established? do you mean that the intervention effect disappeared after the intervention ended? or do you mean that you did not check or could not check (what was the reason?) after the intervention ended ? 

Response 25:  Thanks for pointing this out. It means it cannot be confirmed beyond the research period. We have revised the sentences and made the content clearer.

The maintenance of this form of intervention effect could not be confirmed post research period. (Line 363-364)

Point 26: line 364 ICBT teaches behavioral, emotional, and cognitive control, however observing its potential efficacy in the short term is impractical. your results showed that in the beginning ICBT had a negative effect on anxiety and depression. there was nothing impractical about observing etcetera it was simply not effective 

Response 26: Thanks for the comment. Our results indicated that symptoms of depression and anxiety decreased over time for all three groups. It can be seen at Table 2 and figure 2-3. The ICBT and EW groups exhibited a significant but small reduction in anxiety and depression at T2 and T3 compared to the wait-list control group. The EW group exhibited significantly greater reduction in anxiety and depression than the ICBT group at T2. However, the ICBT group demonstrated greater improvements in alleviating anxiety and depression at T3 compared to the EW group.  

Point 27: line 367 Over the long term, individuals implementing the skills necessary to reduce their anxiety, depression, ICBT was starting to take effect between T2 and T3; or in other words from six weeks to 18 weeks; one of the conclusions of one of your references (the Cochrane review of therapy for MUS) was that trials should conduct follow-up assessments until at least one year after the end of treatment. That would be longer term follow-up; 18 weeks is not long-term it is simply 12 weeks after the end of treatment 

Response 27: Thanks for pointing this out. We have revised the sentences and made the content clearer.

Individuals implementing the skills necessary to reduce their anxiety, depression, and other related emotions, becomes apparent with observable and measurable effect over time. (Line 372-374)

Point 28: line 371 This research was limited in its scope due to several factors: Our sample consists of a relatively homogeneous population with respect to demographic characteristics and educational background, and it may limit the generalizability of the research findings. it wasn't a relatively homogeneous population it was a very homogeneous population as it only included young female nurse students. 

Response 28: Thanks for the comments. We have revised the sentence as suggested.

Our sample consists of a homogeneous population with respect to demographic characteristics and educational background, and it may limit the generalizability of the research findings. Replication in more diverse clinical or community samples is required (Line 376-379).

Point 29: line 373 The exclusion criteria ( such as other type of gastrointestinal diseases diagnosis or receiving CBT within 3 months) used in this study also raised some concerns regarding results generalizability. these two exclusion criteria do not raise concerns about generalisability; it is common sense to exclude those two.

Response 29: Thank you for your helpful suggestion. We have revised the relevant section and removed the sentence “ The exclusion criteria…some concerns regarding results generalizability.”

Point 30: line 377  The randomized control trials we used in the study, though overcome the possible contamination effects, which is likely to occur by contact among individual participants in different groups, it can be susceptible to some methodological problems, such as the number of participants imbalance between treatment groups. first of all this sentence is way too long and secondly it doesn't make any sense. randomisation does not overcome the possible contamination effects even more so because it was an unblinded trial using subjective outcomes.

Response 30: Thanks for the comments. The key word was missing in the previous version. We have changed randomized control trails to “cluster randomized control trails”.

As we discussed in the previous reply, the un-blinded trial of this study was one of the study limitations of this study. It can be seen at line 386-387 “Inadequate participant blinding is another limitation which may result in placebo or nocebo effects on treatment outcomes.

We also addressed the problem of using self-report measures in the section of limitation.

Limitation of this study also include that data on IBS, depression, and anxiety were

derived from self-report questionnaires. The use of multimethod examination for symptom assessment would allow for stronger conclusions. (Line 388-390)

Point 31: line 382  Due to the length of the study, you need to change the start of the sentence because this way it doesn't make any sense why would it be more useful to increase frequency of interventions and extend the follow-up? this is the section about study limitations; so why is the frequency of interventions and the length of the study a limitation? why don't you mention that a no treatment control group is a limitation and the same applies to subjective outcomes in an unblinded trial 
 another limitation is that the patients in the no treatment group had much higher IBS scores and therefore were much more ill 

Response 31Thanks for the comments. We removed the last part of the study limitations as suggested.

As we discussed in our previous reply, in our study, we made the comparison between three groups, they ICBT, wait list control, and expressive writing. The expressive writing served as an active control since research indicated that expressive is useful in reducing anxiety for emotionally expressive IBS patients. We also mentioned in the Introduction that the effects of expressive writing to decrease anxiety and related emotional reactions has been confirmed. In 2.4 intervention, we replace EW by EW active control group.(Line 197)

The problem of unevenly distribution of participants in different groups was also addressed in the study limitations. As it can be seen at line 383-384” such as the number of participants imbalance between wait-list control and treatment groups

The three groups were significantly different at baseline on BSSS, with participants in the waitlist control had higher BSSS scores. However, in the subsequent analyses of each outcome variables, the effect of baseline was controlled for in the GEE and ANCOVA model. (it can be seen at Line 249)

Point 32: a major limitation of the study was the use of a waitlist control group and the fact that the groups were poorly matched as shown by the unevenly distribution of participants (48, 42 and 70) also that (227/394) participants were excluded from the study 

Response 32:Thanks for pointing this out. The problems of using wait-list control group and unevenly distribution of participants have been replied in Response 31.

The target sampling of this study consisted of 394 nursing students, among which, 23 were male, 47 expressed no interest to participate, and 3 dropped out. It’s 73 out of 394 who did not participate before the study began. Then a total of 321 female nursing students completed the IBS screening questionnaire, among which, 135 students (34% of the target sampling) were excluded because their scores did not meet the Rome III criteria. Of the remaining 186 students, 16 were excluded through further diagnostic interview for the reasons not fulfilling Rome III criteria or with history of bowel disease. 

Point 33: line 386 ICBT can effectively reduce anxiety and depression in young women  you can't actually say that based on your study because your study was not done on young women 
 please change that to ICBT can reduce symptoms of anxiety and depression in young Taiwanese female nursing students with mild symptoms 

Response 33

Thanks for pointing this out. In the revision, the sentence has been revised as suggested.

ICBT can reduce symptoms of anxiety and depression in young Taiwanese female nursing students with mild symptoms (line 392-393)

Reviewer 2 Report

I thank the authors for the attention they paid to my questions and suggestions. I still have some minor comments: 

- In l. 64 please ommit the s for ACGS

- The abbreviations IBS-C/ IBS-D (standing for constipation and diarrhea) were not previously defined (l. 67)

- in l. 126-127, the sentence contains no verb "a total of 160 students participant the 127 study (Figure 1)." The authors would want to substitute 'participant' for 'participated in' . 

- in l. 159, the authors report  good internal consistency (Cronbach’s α= 0.95) of the BSSS version. Please speficify at which timepoint this was assessed. 

- In l.159, the authors also report an adequate test–retest of their employed BSSS version (assessed at T0 and T1). Was this studied only in the control group (wait-list group)? If not, this could constitute a limitation since studying test-retest in the whole cohort would be inappropriate since T1 represents two weeks after initiating EW or ICBT. Please explain. 

- In l. 181, please add a space before and a period after the reference "via the internet[37]"

- in l. 252, the authors may want to ommit with : "Therefore, following with the GEE"

- in l. 324, please add a space after the period: "in IBS symptoms.The "

- in l. 348, please substitute "the bottom flow" for "the bottom-up flow"

Author Response

Response to Reviewer 2 Comments
Thank you for your very careful review of our manuscripts, and for the comments,
corrections and suggestions that ensued. Your comments helped us refine our paper by
clarifying critical sections and presenting results. Please refer to the manuscript for
numerous changes motivated by your comments. Below we summarize major
comments you provided and our reaction to them.
Point 1: In l. 64 please ommit the s for ACGS
Response 1: Thanks for pointing this out. Modified as suggested. (Line 67)

Point 2: The abbreviations IBS-C/ IBS-D (standing for constipation and diarrhea)

were not previously defined (l. 67)
Response 2: Thanks for pointing this out. Modified as suggested. (Line 70)
Point 3: in l. 126-127, the sentence contains no verb "a total of 160
students participant the 127 study (Figure 1)." The authors would want to substitute
'participant' for 'participated in'.
Response 3: Thanks for pointing this out. Modified as suggested. (Line 130)
Point 4: in l. 159, the authors report good internal consistency (Cronbach’s α= 0.95)
of the BSSS version. Please specify at which time-point this was assessed.
Response 4: Thanks for the suggestion. We have added time points and number of
sample for the measurement.
BSSS demonstrated good internal consistency (Cronbach’s α= 0.95 at T0, N = 160 )
(Line 161-162)
Point 5: In l.159, the authors also report an adequate test–retest of their employed
BSSS version (assessed at T0 and T1). Was this studied only in the control group
(wait-list group)? If not, this could constitute a limitation since studying test-retest in
the whole cohort would be inappropriate since T1 represents two weeks after
initiating EW or ICBT. Please explain. Limitation
Response 5: Thanks for the thoughtful comments. We used the data of want-list
control group for BSSS test-retest analysis. We have added more information in that
sentence.
…and adequate test–retest reliability (r = 0.76, p<0 .001, assessed at T0 and T1
for waitlist control group, N = 70).(Line 162-163)
Point 6: In l. 181, please add a space before and a period after the reference "via the
internet[37]"
Response 6: Thanks for pointing this out. Modified as suggested. (Line 183)
Point 7: in l. 252, the authors may want to ommit with : "Therefore,
following with the GEE"
Response 7: Thanks for pointing this out. Modified as suggested. (Line 254)
Pont 8: in l. 324, please add a space after the period: "in IBS symptoms.The "
Response 8: Thanks for pointing this out. Modified as suggested. (Line 326)
Point 9: in l. 348, please substitute "the bottom flow" for "the bottom-up flow"
Response 9: Thanks for pointing this out. Modified as suggested. (Line 353)

Round  2

Reviewer 1 Report

 I would suggest  publishing the manuscript if the authors make a few small adjustments as outlined below

143

please change 200-100 words within 6 weeks.

 To

100-200 words within 6 weeks.

 figure 1

 On the right there are boxes explaining what happens too but not for the following:

Recruitment (321)  to Enrollment (186)   so please provide a box there too

 196

Assessments were held on the 2nd, 6th, and 18th week of the study

please change that to something like:

Assessments were held at 2 weeks (during therapy), 6 weeks (end of therapy) and 18 weeks after the start of treatment.

 204 and 207

 same as in 196 it is not on the 2nd, 6th, and 18th week of the study

 but  at 2, 6 and 18 weeks

Author Response

Response to Reviewer 1 Comments

Thank you for your very careful review of our manuscripts, and for the corrections and suggestions that ensued. The responses are given as below:

Point 1: 143 please change 200-100 words within 6 weeks, to 100-200 words within 6 weeks.

 Reponse 1: Thanks for pointing this out. Modified as suggested (Line 143) .

Point 2:  figure 1, on the right there are boxes explaining what happens too but not for the following: Recruitment (321)  to Enrollment (186)   so please provide a box there too

Reponse 2: Thanks for the suggestion. The figure has been modified as suggested. (page.4, figure 1) 

Point 3: 196, Assessments were held on the 2nd, 6th, and 18th week of the study

please change that to something like: Assessments were held at 2 weeks (during therapy), 6 weeks (end of therapy) and 18 weeks after the start of treatment.

 Response 3: Thanks for pointing this out. The sentence has been modified as suggested. (Line 197-198)

 Point 4: 204 and 207, same as in 196 it is not on the 2nd, 6th, and 18th week of the study but  at 2, 6 and 18 weeks

Response 4: Thanks for the suggestion. Modified as suggested. (Line 205 and 208).

This manuscript is a resubmission of an earlier submission. The following is a list of the peer review reports and author responses from that submission.

Round  1

Reviewer 1 Report

General comments :

-        It is an interesting manuscript that evaluates the effectiveness of internet-based cognitive behavior therapy (CBT) compared to expressive writing or a no-treatment group in young nursing female students with irritable bowel syndrome. The manuscript reports positives CBT outcomes and is worth publishing. However, several issues require to be revised.

-        Please review the manuscript for excessive spacing.

-        Although the English is comprehensible, some sentences are difficult to read and the manuscript would benefit from proofreading by a native speaker.

-        A major limitation is the absence of validation of this scale in Chinese. The absence of published psychometric properties (e.g., test-retest reliability) on BSSS does not allow drawing formal conclusions on the protocol results. The authors are advised to add more information on the translation process (e.g. forward backward translation?). In addition, this limitation needs to be addressed in the discussion.

Abstract :

-        Please add a space in the following : T3compared

-        Some abbreviations could be omitted since the word did not recur later: CBT, BSSS, CES-D, STA-I

-        In line 23, please add a period after ‘the control group’

-        Although it might look logical to use T1, T2, and T3 for the three points in time. These points were never defined and might confuse readers.

Introduction:

-        The introduction could benefit from adding more details on the symptomatology of IBS.

-        In l.42 please remove the semicolon (;) after such as

-        Some sentences lack references.

o   In l. 49 citing the British Society of Gastroenterology.

o   In l. 59-61 citing the effectiveness of internet based CBT in several pathologies

-        In l.56 please substitute ‘dependent’ for ‘depending’

-        In l. 76, the abbreviation ICBT was already defined previously.

-        It would be interesting to mention the rationale for testing the efficacy of internet-based CBT. For an example, please check the following reference: Andersson E, Ljótsson B, Smit F, Paxling B, Hedman E, Lindefors N, Andersson G, Rück C. Cost-effectiveness of internet-based cognitive behavior therapy for irritable bowel syndrome: results from a randomized controlled trial. BMC Public Health. 2011 Apr 7;11:215. doi: 10.1186/1471-2458-11-215.

Methods:

-        Please cite a reference for Rome III criteria.

-        Please specify the age range for inclusion

-        It would be helpful for readers from countries that have different nursing programs to understand the practicum in Taiwan. Adding a sentence would be helpful.

-        No criteria concerned the pharmacological therapies. Was any participant receiving any treatment? In the limitation section, the reader will discover that the authors did not include patients with pharmacological therapies for anxiety or depressions. This should be moved to methods section, and the authors need to tell whether the participants received drugs for IBS or other conditions.

-        In the method, the number of participants in the EW group is 48. In figure 1, it is 42. Please clarify.

-        It is important to provide more information on the contents of ICBT sessions.

-        Please specify the evaluation points for the EW group.

-        Regarding statistical analysis, the authors used parametric tests (i.e., ANOVA). It is important to mention if the data respected the assumptions for conducting ANOVA/ANCOVA (i.e., normality of distribution, homogeneity of variances, etc.).

Figure 1.

-        First, the figure is incomplete. Please resubmit the file

-        Please correct ‘quite school’. Did the authors mean ‘quit’ as for ‘dropped out’?

-        In the outcome, please correct CED-S for CES-D

Results:

-        The main concern is the way the results are displayed which could be misleading. The authors state the mean age of all participants rather than the mean age for each group. This should be clarified to demonstrate any potential group difference with regard to age.

-        Again, the academic years of participants could be clarified. In the method, the authors mention that they selected the students that are about to start their practicum. In the results, one can understand that the participants were in their fourth year and third year of a 5-year junior college program. Please clarify.

-        No need to restate Rome III criteria here (l.185) since it is already mentioned in the method.

-        In l.186-189, the sentence should be relocated in the method since it does not concern the results but rather the inclusion criteria: “Participants had to meet the following criteria: (1) experience of repeated abdominal discomfort or pain at least 3 days per month within the preceding 3 months; and (2) experience at least two of the following: symptomatic relief after defecation, change in defecation frequency, and change in the hardness of feces.”

-        In l. 189-191, there is an error in calculation since the sum does not give 157 and not 160: “A total of 160 participants (41.1%) fulfilled the diagnosis criteria: 66 participants (41.3%) were constipation predominant, 63 (39.4%) were mixed predominant, and 28 (17.5%) were diarrhea predominant.”

-        In l.196-197 and in l. 200-201, please correct the structure of the sentence: “These three mean scores differed no significantly”.

-        BSSS scores differed significantly between the three groups. Please add the p-value here.

Discussion:

-        The discussion is brief and could be more developed.

-        The following statement applies for the anxiety and depression symptoms but not for IBS symptoms which were more relieved at T2: “Greater differences between ICBT and the other two methods were observable in the long term, particularly on the 18th week, where ICBT exhibited notable efficacy.”  The discussion may benefit from a more thorough interpretation of CBT effects on IBS symptoms. In other words, the authors could explain why only CBT had significant effects on IBS symptoms and why these effects peaked at T2. This might be related to the number and duration of sessions which could be solved in future by increasing CBT ‘dose’?

-        It would be interesting to enrich the discussion by explain the results in the light of the current knowledge on CBT mechanisms of action:

o   CBT appears to exert its effects on anxiety and depression by enhancing the top-down cognitive control of emotions (e.g. prefrontal regions) and/or decreasing the bottom flow of fear-related emotions from limbic structures

§  Reference for anxiety: Brooks SJ, Stein DJ. A systematic review of the neural bases of psychotherapy for anxiety and related disorders. Dialogues Clin Neurosci. 2015 Sep; 17(3): 261–279.

§  References for depression: Chalah MA, Ayache SS. Disentangling the Neural Basis of Cognitive Behavioral Therapy in Psychiatric Disorders: A Focus on Depression. Brain Sci. 2018 Aug 9;8(8). pii: E150.

o   CBT effects on IBS could be speculated/hypothesized from studies showing  increased activity in the insula and reduced activation of the dorsolateral prefrontal cortex (DLPFC) in response to visceral stimulation in patients with IBS:

§  Aizawa E, Sato Y, Kochiyama T, Saito N, Izumiyama M, Morishita J, Kanazawa M, Shima K, Mushiake H, Hongo M, Fukudo S. Altered cognitive function of prefrontal cortex during error feedback in patients with irritable bowel syndrome, based on FMRI and dynamic causal modeling. Gastroenterology. 2012 Nov;143(5):1188-1198.

Author Response

Thanks for careful review. The revised manuscript benefited greatly by the insightful suggestions and critical comments proposed by the reviewer. The responses are given as below:

General comments:

 It is an interesting manuscript that evaluates the effectiveness of internet-based cognitive behavior therapy (CBT) compared to expressive writing or a no-treatment group in young nursing female students with irritable bowel syndrome. The manuscript reports positives CBT outcomes and is worth publishing. However, several issues require to be revised.

Remarks

(1)   Please review the manuscript for excessive spacing.

Reply (1): Thanks for pointing this out, a number of excessive spacing had been

modified.

(2)   Although the English is comprehensible, some sentences are difficult to read and the manuscript would benefit from proofreading by a native speaker.

Reply (2): Thanks for the suggestion. A native English speaker who is graduate student in medical science has helped proofread the revised manuscript.

(3)   A major limitation is the absence of validation of this scale in Chinese. The absence of published psychometric properties (e.g., test-retest reliability) on BSSS does not allow drawing formal conclusions on the protocol results. The authors are advised to add more information on the translation process (e.g. forward backward translation?). In addition, this limitation needs to be addressed in the discussion.

Reply (3): thanks for point this out. We have added the following translation process

in the first paragraph of Measure.

The three-procedure approach suggested by Brislin (1970) was adopted for translation. The English inventories were first translated by both authors into Chinese, and the Chinese inventory was then back-translated into English by a native English speaker. The authors compared the back-translation version with the original and revised the Chinese inventories as required. Three graduate students were asked to fill out both Chinese and English inventories, and further revision was made when inconsistent responses were found between the two versions of the same inventory. (Line 152-158)

As for the psychometric properties of Chinese version of BSSS, we reported internal consistency Cronbach’s α and test-retest reliability of the current study.  

In the current study, BSSS demonstrated good internal consistency (Cronbach’s α= 0.95) and adequate test–retest (assessed at T0 and T1) reliability (r = 0.75, p << span="">0 .001). (Line 158-160)

Abstract :

(4) Please add a space in the following : T3compared

Reply (4) modified as suggested. (Line 26)

(5)Some abbreviations could be omitted since the word did not recur later: CBT,

 BSSS, CES-D, STA-I

 Reply (5) Thanks for point this out. The abbreviations had been deleted.

(6) In line 23, please add a period after ‘the control group’

Reply (6) Modified as suggested. (Line 23-24)

(7) Although it might look logical to use T1, T2, and T3 for the three points in time.

   These points were never defined and might confuse readers.

Reply (7) Revised as suggested. We have added more information in the abstract.

The treatment interventions lasted for 6- weeks. The level of anxiety , depression and IBS symptoms were assessed at four points in time, baseline assessment at T0, and 2 weeks later (T1) and the end of practicum (T2) , and at 3 month follow-up (T3). (Line 19-22)

 Introduction:

(8) The introduction could benefit from adding more details on the symptomatology

   of IBS.

Reply (8) We have added more symptoms of IBS as suggested.

Such as defecation-related abdominal discomfort and pain, bowel dysfunction, and abdominal bloating in the absence of any structural abnormality (Line 33-34)

(9) In l.42 please remove the semicolon (;) after such as

Reply (9) Corrected as suggested.

(10) Some sentences lack references.

       In l. 49 citing the British Society of Gastroenterology. In l. 59-61 citing the effectiveness of internet based CBT in several pathologies

Reply (10) we have added citation to the two sentences.

The British Society of Gastroenterology guidelines on the IBS mechanism and practical management concluded that more accurate means of identifying IBS-C/ IBS-D are required in order to provide the most appropriate treatment[12]. (Line 65-68)

Internet-delivered CBT (ICBT) has been used to treat over 100 types of syndromes and 20 clinical diseases exhibiting treatment results comparable to those of conventional CBT [20]. (Line 81-82)

(11)  In l.56 please substitute ‘dependent’ for ‘depending’

Reply (11) Modified as suggested. (Line 78)

(12)In l. 76, the abbreviation ICBT was already defined previously.

Reply (12) Corrected as suggested.

(13) It would be interesting to mention the rationale for testing the efficacy of internet-based CBT.

Reply (13) We added the rationale for conducting the cluster randomized control trails on ICBT for female adolescents.

A majority of the evaluated ICBT programs for IBS are for clinical patients [20], to our knowledge this is the first study to examine ICBT for female students with IBS. (Line 104-106)

 Methods:

(14) Please cite a reference for Rome III criteria.

Reply (14) added citation as suggested.

diagnosis by a general practitioner meeting the Rome III criteria [30]. (Line 113)

(15)Please specify the age range for inclusion

Reply (15) Thanks for pointing it out. We have added the age range of the participants.

A total of 321 participants (18 to 22 years ) completed the IBS screening questionnaire, (Line 120)

(16)It would be helpful for readers from countries that have different nursing

programs to understand the practicum in Taiwan. Adding a sentence would be helpful.

Reply (16) We have added more information introducing nursing practicum in Taiwan.

In Taiwan, a five-year nursing college program requires students to take several short-term practicum courses beginning in the third year toward the end of their fifth year. Nursing students are divided into groups when they are assigned to practicum sites. (Line 129-132)

(17) No criteria concerned the pharmacological therapies. Was any participant

receiving any treatment? In the limitation section, the reader will discover that

the authors did not include patients with pharmacological therapies for anxiety or

depressions. This should be moved to methods section, and the authors need to

tell whether the participants received drugs for IBS or other conditions.

Reply (17) One of the exclusion criteria for the participation was receiving CBT in the preceding 3 months. We did not exclude students for participation if they were taking medication for IBS related problems. However, we check that in our questionnaire, and found none of the participants reported to take medication for their IBS related problems. We added more information in the results.

None of the participants reported to take medication for their IBS related problems. (Line 233-234)

(18)In the method, the number of participants in the EW group is 48. In figure 1, it is

42. Please clarify.

  Reply (18) Typo had been modified. It should be 42.

 There were 42 participants in the EW group which required…(Line 136)

(19)It is important to provide more information on the contents of ICBT sessions.

 Reply (19) As suggested, the 2.4. Intervention is replaced by  

ICBT group: the ICBT intervention was developed by researchers based on Mind Over Mood  and made available to the study participants via the internet[35] The protocol consists of 13 sessions and encompasses behavioral, emotional, and cognitive components of stress management. The 13 intervention sessions are divided into three parts. Part one: sessions 1–4 focus on behavioral strategies and abdominal breathing and progressive muscle relaxation training. Part two: Sessions 5–7 focus on emotional strategies and performing and recording pleasant activities. Part three: Sessions 8–13 focus on cognitive strategies to teach the participants how to recognize negative thoughts and record the recognition process. (Line 180-187)

(20) Please specify the evaluation points for the EW group.

 Reply (20) We have added more details on EW group.  

The EW assignments were uploaded to the system to check for compliance with the intervention, but no further evaluation of content was employed. Time point of the outcome assessment was the same as ICBT group. (Line 199-201)

(21) Regarding statistical analysis, the authors used parametric tests (i.e., ANOVA).

It is important to mention if the data respected the assumptions for conducting

ANOVA/ANCOVA (i.e., normality of distribution, homogeneity of variances,

etc.).

  Reply (21) Thanks for pointing this out. We have added whether the

assumptions of ANOVA/ANCOVA were met.

 The assumptions of normality of the distribution of the data and homogeneity of

 variances were confirmed by the Kolmogorov-Smirnov test and the Levene test of

 equal variances. (Line 221-223)

Figure 1.

(22) First, the figure is incomplete. Please resubmit the file

 Reply (22) Figure 1 has been revised.   

(23)Please correct ‘quite school’. Did the authors mean ‘quit’ as for ‘dropped out’?

Reply (23) It was a tyop. Quite has been replaced by dropped out.

(24) In the outcome, please correct CED-S for CES-D

Reply (24) Typo was corrected.

 Results:

(25)The main concern is the way the results are displayed which could be

    misleading. The authors state the mean age of all participants rather than the

    mean age for each group. This should be clarified to demonstrate any potential

    group difference with regard to age.

Reply (25) Thanks for pointing this out. The mean age for each group was added as suggested.

The ICBT group was 19.27 ± 0.45 years, the EW group was 19.45 ± 0.55 years, and the wait-list control group was 18.47 ± 0.70 years. (Line 228-229)

(26)Again, the academic years of participants could be clarified. In the method, the

    authors mention that they selected the students that are about to start their

    practicum. In the results, one can understand that the participants were in their

    fourth year and third year of a 5-year junior college program. Please clarify.

Reply (26) As mentioned in remark # (16), we have added more information to describe the nursing program in Taiwan. A five-year nursing college program requires students to take several short-term practicum courses beginning in the third year toward the end of their fifth year.

(27) No need to restate Rome III criteria here (l.185) since it is already mentioned in  

    the method.

Reply (27) Deleted as suggested.

(28) In l.186-189, the sentence should be relocated in the method since it does not

    concern the results but rather the inclusion criteria: “Participants had to meet the

    following criteria: (1) experience of repeated abdominal discomfort or pain at

    least 3 days per month within the preceding 3 months; and (2) experience at least

    two of the following: symptomatic relief after defecation, change in defecation

    frequency, and change in the hardness of feces.”

Reply (28) Indeed. It’s redundant. The criteria were stated in the Materials and

Methods. We deleted it as suggested.  

(29) In l. 189-191, there is an error in calculation since the sum does not give 157 and

    not 160: “A total of 160 participants (41.1%) fulfilled the diagnosis criteria: 66

    participants (41.3%) were constipation predominant, 63 (39.4%) were mixed

    predominant, and 28 (17.5%) were diarrhea predominant.”

Reply (29) Thanks for pointing this out. We have re-calculated and corrected the numbers of participants in each category of the IBS.

A total of 160 participants (41.1%) fulfilled the diagnositic criteria for IBS: 67 participants (41.88%) were constipation predominant, 66 (41.25%) were mixed predominant, and 27 (16.87%) were diarrhea predominant. (Line 231-233)

(30)  In l.196-197 and in l. 200-201, please correct the structure of the sentence:

     “These three mean scores differed no significantly”.

Reply (30) Revised as suggested.  

These three mean scores showed no significant difference (Line 240 and Line 244)

(31) BSSS scores differed significantly between the three groups. Please add the

    p-value here.

Reply (31) we have added p-value for significant differences between groups for BSSS at baseline.

 These three mean scores differed significantly (F=10.90, p<0.001). (Line 237)

Discussion:

(32) The discussion is brief and could be more developed.

Reply (32) . We have revised the text to expand on the discussion of the non-significant differences on IBS between the three groups, and the effects of ICBT on depression and anxiety at follow-up. Please see the two new paragraphs as below.

The cluster randomized controlled trail was conducted to compare the effects of internet delivered cognitive therapy (ICBT) with self-administered expressive writing and wait-list control on IBS and its related emotional symptoms. The results indicated that symptoms of IBS as well as depression and anxiety decreased over time for all three groups with only one exception, the IBS scores increased slightly at the end of the treatment for the ICBT group. Contrary to our hypothesis, no significant differences were observed between the three groups, indicating that neither ICBT nor EW was superior to the wait-list control to ease IBS symptoms. This finding is similar to the conclusion of a recent meta-analysis [36], in which CBT delivered through the internet, or minimal contact CBT showed no improvement in IBS symptoms.The authors speculated that direct contact between therapists and patients/clients was necessary for psychological therapy to be effective. A number of controlled trial studies demonstrated CBT among other personal contact psychological therapies were beneficial for alleviating symptoms[37,38],. Another explaination is the length of time of the interevention or even the number of psychoanalytic sessions attended by the IBS sufferer. CBT requires consistent guided effort and focus, of which 8-10 weeks is insufficient time to accurately assess the potential for IBS. (Line 313-330)

ICBT can effectively treat anxiety and depression but not IBS symptoms. The results of this study are similar to the Boyce study [17] which supports that the proposed cognitive model appears to have its effect by altering the cognitive response to visceral hypersensitivity. The Cochrane review of non-pharmacological interventions for functional syndromes (14 of the 21 RCTs used CBT) showed excellent effect when CBT was learned by the participants [39]. It is known, that at least half of IBS patients suffer from anxiety, depression or some form of hypochondriacal manifestation of mental illness. Studies from terriary care suggested that more than 60% of IBS sufferers were in fact suffering from other psychiatric disorders- most commonly anxiety or depressive disorder [40,41,42].The cognitive mechanism of effect between CBT and the emotional symptoms of anxiety and depression are currently being researched. Understanding the underpinnings of the possibility that the mind affects the physical experience and health within an individual suggests strongly, that CBT, when it is also similarly refined in its practice, may prove to be a powerful tool in the health care arsenal for both physican and patient. (Line 331-343)

(33) The following statement applies for the anxiety and depression symptoms but not   

    for IBS symptoms which were more relieved at T2:

    “Greater differences between ICBT and the other two methods were observable

     in the long term, particularly on the 18th week, where ICBT exhibited notable

     efficacy.” 

Reply (33) Indeed. The treatment effects of ICBT on IBS are not significant. We have modified the statement in the first paragraph of discussion, which also can be seen in our reply to remark # (32).

(34) The discussion may benefit from a more thorough interpretation of CBT effects

    on IBS symptoms. In other words, the authors could explain why only CBT had

    significant effects on IBS symptoms and why these effects peaked at T2.4This

    might be related to the number and duration of sessions which could be solved in

    future by increasing CBT ‘dose’?

Reply (34) We have added some information to explain the possible reason for the increase of IBS scores at Time 2 for ICBT.  

The results indicated that symptoms of IBS as well as depression and anxiety decreased over time for all three groups with only one exception, the IBS scores increased slightly at the end of the treatment for the ICBT group. We speculated that self-help learning of ICBT may become additional stress while students usually had a great deal of assignments need to be completed shortly after the end of practicum. This may explain why the scores of IBS increased for ICBT group at T2. (Line 313-320)

 (35)It would be interesting to enrich the discussion by explain the results in the light

   of the current knowledge on CBT mechanisms of action:

   CBT appears to exert its effects on anxiety and depression by enhancing the

   top-down cognitive control of emotions (e.g. prefrontal regions) and/or decreasing

   the bottom flow of fear-related emotions from limbic structures

   Reference for anxiety: Brooks SJ, Stein DJ. A systematic review of the neural

   bases of psychotherapy for anxiety and related disorders. Dialogues Clin  

   Neurosci. 2015 Sep; 17(3): 261–279.

    References for depression: Chalah MA, Ayache SS. Disentangling the Neural

   Basis of Cognitive Behavioral Therapy in Psychiatric Disorders: A Focus on

   Depression. Brain Sci. 2018 Aug 9;8(8). pii: E150. 

    CBT effects on IBS could be speculated/hypothesized from studies

   showing  increased activity in the insula and reduced activation of the

   dorsolateral prefrontal cortex (DLPFC) in response to visceral stimulation in

   patients with IBS.

   Aizawa E, Sato Y, Kochiyama T, Saito N, Izumiyama M, Morishita J, Kanazawa

   M, Shima K, Mushiake H, Hongo M, Fukudo S. Altered cognitive function of

    prefrontal cortex during error feedback in patients with irritable bowel syndrome,    

    based on FMRI and dynamic causal modeling. Gastroenterology. 2012     

    Nov;143(5):1188-1198.

Reply (35) Thanks for the very thoughtful suggestions. We have added one new  

paragraph to address the mechanism of cognitive behavioral treatment on emotions.

Greater differences between ICBT and the other two methods were observable particularly at the 18th week, where ICBT exhibited notable efficacy. The results of this study are in agreement with the results of previous studies [45, 46, 47, 48]. ICBT appears to exert its effects on anxiety and depression by enhancing the top-down cognitive control of emotions (e.g. prefrontal regions) and/or decreasing the bottom flow of fear-related emotions from limbic structures. Cognitive-behavioral control appears to lead to increased awareness, improved emotional health and con-comittant physical health over the long term. (Line 344-350)

Reviewer 2 Report

Thank you for asking me to Review this interesting study.

 There are a number of little things and issues with this article. 

Review report 

 a brief summary 

the authors tested the efficacy of Internet-based CBT and expressive writing in female nursing students in Taiwan with IBS. it is an interesting paper but unfortunately they ignore their own null effect as well as a number of weaknesses of the study 

 Abstract 

Control group please change this to waitlist control group.

 Please make it clear what T2 and T3 are. 

 Introduction 

 The Rome III criteria have been replaced by Rome IV criteria in early 2016.

 As noted by a Review by Weaver in 2017, "research has shed light on IBS pathophysiology, therapeutic interventions remain symptom driven, employing both pharmacologic and nonpharmacologic approaches. "

Weaver (2017) noted that In 40% to 60% of cases, IBS is accompanied by depression or anxiety. This also means that in the other 50% it is not associated with psychological aspects.

Weaver (2017) also noted that there is Evidence of biologic dysregulation but the exact mechanisms leading to IBS symptoms are not completely understood. Which the authors of this article ignore. Curative medical interventions have yet to be discovered, therefore treatment focuses on reducing symptoms. There are a number of different pharmacological interventions but also some behavioural ones. 

It is important that this is made clearer by the authors and that not a whole introduction - apart from one sentence - is devoted to stress the psychological nature of this disease which is incorrect. CBT should simply be presented as one of the possible symptomatic treatments which (just like multicomponent psychological therapy, dynamic psychotherapy, and hypnotherapy) is only mildly beneficial as found by a Cochrane review 2009) and a systematic review by Ford (2014) which included 30 studies on the effect of psychological therapies on patients with IBS.

Weaver, Kristen Ronn MS, ACNP, ANP; Melkus, Gail D'Eramo EdD, C-NP, FAAN; Henderson, Wendy A. PhD, MSN, CRNP Irritable Bowel Syndrome AJN, American Journal of Nursing: June 2017 - Volume 117 - Issue 6 - p 48–55

doi: 10.1097/01.NAJ.0000520253.57459.01

Moreover, the American College of Gastroenterology ( 2014 ) performed a systematic review to determine the efficacy of pharmacologic and nonpharmacologic IBS therapies. Only two therapies overall received strong recommendations for use and were supported, respectively, by evidence of high and moderate quality: linaclotide and lubiprostone for the treatment of IBS-C. 

 Page 2 line 51/52 

 relaxation training and CBT are not among the most efficient methods; secondly the authors probably mean the most effective and not the most efficient 

 I think it's important that the tone of the introduction is changed; at the moment it is very women unfriendly; the authors are blaming women for their IBS; also the authors seem to confuse correlation and causation. 

The authors mention the review by the Laird; this review advised to use active control conditions to control for non-specific treatment effects. Unfortunately this study didn't do that but used a waitlist control group. 

The authors fail to mention the problems of a waitlist control group. this is of particular importance because knowledge that one is not receiving treatment, affects outcomes negatively. Assignment to no treatment may strengthen participants' beliefs that they will not improve, thereby reducing the chance of spontaneous improvement. Participants randomized to waitlist control conditions may improve less than would be expected compared to participants not enrolled in a trial. Using waitlist or no treatment control conditions can lead to the overestimation of the effectiveness of a treatment (Mohr et al., 2009).

A Cochrane review of non-pharmacological interventions for functional syndromes (14 of the 21 RCTs used CBT) (Van Dessel et al., 2014), and a recent meta-analysis of 33 RCTs of mindfulness-based interventions (Dunning et al., 2018), both not only concluded that most benefits disappear when an active control group is used, instead of a waitlist or no treatment control group. But also, that there's evidence too that the remaining effects are inflated by bias and multiple methodological concerns, including high drop-out rates and selective biases in sampling.

 Page 3 line 100

 the study uses a Cluster randomised controlled trial and there are some pitfalls of this sort of trial; for example a reduced statistical efficiency relative to randomised controlled trials with the same number of individuals. what has the study done to address this? 

 Page 3 line 102

 eligible students were randomly assigned to 1 of three groups but how can it then be that in the two treatment groups there were 48 and 42 participants but in the control group there were 70 ? if the randomisation had been done properly it should have been the same number in each group 

 Page four figure 1

 at the bottom of the figure please explain what T0, T1, T2 and T3 stand for.

 page five line 125

 if the scale was translated by the authors into Chinese this at the same time means that the Chinese version of BSSS is not validated

 line 155 

 outcome assessments were held on the 2nd, 6th 18th week 

 what does this mean? where they held during the second week or two weeks after starting treatment and the same applies to 6 and 18 weeks 

 page 6 line 185

The Rome III criteria was used as diagnostic criteria

 please change was to were because criteria is plural

 as from the beginning of 2016 the Rome IV criteria are being used then please add when your trial started and patients were recruited because otherwise your trial was outdated before it started.

 Page 6 line 192 to 194

 the BSSS analysis scores are the following 

24.96 ICBT group, 19.07 EW group and 32.00 control group; the differences are statistically significant; this basically means that there was a significant difference in severity between the ICBT group and the control group of 30% (7.04/24.96) and the difference with the EW group was almost double that. therefore the participants in the control group had more severe IBS, this means that the groups were poorly matched 

 page 6

197 and 201 differed no significantly 

 should be differed not significantly 

 Page six line 208

 it is misleading to say that the ICBT group and EW group exhibited considerable improvements compared with the control group and then add but the difference was not significant. this sentence should be rewritten into something like there was no statistically significant improvement after ICBT (p=0.341) and EW (p=0.373) compared to the control group at T1 (2 weeks after the start of treatment). please leave the β values away. 

 the sentences about the improvements after six and 18 weeks should also be changed in a similar way. 

 At T2 and at T3 there were also no statistically significant improvements after ICBT and EW compared to the control group according to the authors themselves. It is not relevant that the ICBT group exhibited a greater change than the control group at 18 weeks because the improvements were not statistically significant. So in other words your study has shown that ICBT and EW are not more effective than the waitlist for IBS symptoms; also known as a null effect. 

 This also means that the abstract, discussion and conclusion need to be changed accordingly.

 As regards to depression and anxiety, it is no surprise that CBT is more effective than leaving people on the waitlist  as a meta analysis by Tolin showed that CBT is the most effective treatment for both problems.

 Table 1

 please explain what intercept, group, time, group time and baseline mean

 what is DF 

 what is bald x2

  I think everybody understands that p<0.001 means p<0.001 so what's the point in adding asterixis after the p values? 

 table 2

 why are there no P values for control?

 what does baseline mean?

 Under T2 why are there no beta entries at baseline?

 what does T1, weeks 2 mean do you mean during the second week? please clearly state what it means

 same applies to T2 and T3 

 Figure 2 is misleading as the BSSS changes at T1, T2 and T3 are not statistically significant compared to the control group.

 Page 10 discussion 

 I thought this was a study about IBS yet suddenly in the first line of discussion it is called gastrointestinal syndrome and not IBS ? please be consistent and and change it to IBS 

 there are no statistically significant differences between ICBT, EW and the control group at T1, T2 and T3 therefore this study has shown that neither ICBT nor EW are an effective treatment for IBS.

 Line 276 

 symptoms were not resolved; they had improved in all three groups with no statistically significant differences between the three groups 

 line 277 

it is not difficult to observe the efficacy of ICBT in the short term for the simple reason that there was no statistically significant difference in improvement between the three groups at any of the three points 

 study limitations 

 there are disadvantages of a cluster randomised controlled trial which should be mentioned here too

 it was a nonblinded study using subjective outcomes; this combination is unreliable and likely to lead to false positive results 

the study used a Chinese version of BSSS yet this was not validated

 it is odd that everybody in the three groups who started the trial also finished it. Were patients super motivated to receive ICBT or were they offered an incentive if they fulfilled all three assessments ?

 a major limitation of the study was not only the use of a waitlist control group but also the fact that patients in the control group had more severe IBS then in the other two groups. This means that the groups were poorly matched. this is also shown by the unevenly distribution of participants (48, 42 and 70). this suggests that there was a major problem with randomisation  

The discussion needs to be rewritten and the conclusion should be that ICBT and EW might reduce symptoms of depression and anxiety but are not effective treatments for IBS 

 page 11 line 288 to 290 

what does this sentence mean? 

 Conclusion 

as mentioned before the conclusion needs to be rewritten 

 and the same applies to the abstract

Rating the manuscript 

 originality/novelty: 

 significance 

 the conclusions by the authors are not supported by their results.

 they ignore their own null effect even though they themselves noted that at T1 T2 and T3 there are no statistically significant differences between treatment and waitlist control group 

quality of presentation 

 the data and analysis are not presented appropriately as they ignore their own null effect 

 scientific soundness 

 see earlier points 

 interest to the readers 

 if they would report their own null effect then it would be an interesting article 

 overall merit

 there is an overall benefit in publishing this work if they make the aforementioned changes

 English level

 in general it's appropriate and understandable but there are a few sentences as mentioned before which they need to change and make it clearer what they mean or explain things better.

 overall recommendation

 reconsider after major revisions

Author Response

Thanks for careful review. The revised manuscript benefited greatly by the insightful suggestions and critical comments proposed by the reviewer. The responses are given as below:

A brief summary 

the authors tested the efficacy of Internet-based CBT and expressive writing in female nursing students in Taiwan with IBS. it is an interesting paper but unfortunately they ignore their own null effect as well as a number of weaknesses of the study 

 Remarks

Abstract 

(1)   Control group please change this to waitlist control group.

Reply (1) Modified as suggested. All “control group” was replaced by “waitlist control group”.

(2)   Please make it clear what T2 and T3 are. 

Reply (2) Thanks for pointing that out. We have revised the abstract as suggested.

The treatment interventions lasted for 6- weeks. The level of anxiety , depression and IBS symptoms were assessed at four points in time, baseline assessment at T0, and 2 weeks later (T1) and the end of practicum (T2) , and at 3 month follow-up (T3). (Line 19-22)

Introduction 

 (3)The Rome III criteria have been replaced by Rome IV criteria in early 2016.

Reply (3) Thanks for pointing this out. We have added some more information regarding the Rome IV.

The Rome III criteria for functional GI disorders served as the symptom-based diagnostic criteria for IBS since its release in 2006 until early 2016, since then, the Rome Foundation has updated the criteria with the release of Rome IV [2]. (Line 35-37)

(1)   As noted by a Review by Weaver in 2017, "research has shed light on IBS pathophysiology, therapeutic interventions remain symptom driven, employing both pharmacologic and nonpharmacologic approaches. " Weaver (2017) noted that In 40% to 60% of cases, IBS is accompanied by depression or anxiety. This also means that in the other 50% it is not associated with psychological aspects.

 Weaver (2017) also noted that there is Evidence of biologic dysregulation but the

exact mechanisms leading to IBS symptoms are not completely understood.

Which the authors of this article ignore. Curative medical interventions have yet

to be discovered, therefore treatment focuses on reducing symptoms. There are a

number of different pharmacological interventions but also some behavioural

ones.It is important that this is made clearer by the authors and that not a whole

introduction - apart from one sentence - is devoted to stress the psychological

nature of this disease which is incorrect. CBT should simply be presented as one

of the possible symptomatic treatments which (just like multicomponent

psychological therapy, dynamic psychotherapy, and hypnotherapy) is only mildly

beneficial as found by a Cochrane review 2009) and a systematic review by Ford

(2014) which included 30 studies on the effect of psychological therapies on

patients with IBS.

  Weaver, Kristen Ronn MS, ACNP, ANP; Melkus, Gail D'Eramo EdD, C-NP, FAAN; Henderson, Wendy A. PhD, MSN, CRNP Irritable Bowel Syndrome AJN, American Journal of Nursing: June 2017 - Volume 117 - Issue 6 - p 48–55

doi: 10.1097/01.NAJ.0000520253.57459.01

Reply (4) Thanks for the thoughtful comments. Indeed, the etiology of IBS remains uncertain. We addressed this issues in the revised manuscript.

Both physiological and psychological variables play key roles in the etiology of IBS and perpetuate symptoms [7]. Evidence of biological dysregulation has been reported in patients with IBS and efforts to understand the neurohormonal underpinnings of the disorder are ongoing, however the exact mechanisms leading to IBS symptoms are not completely understood [8] (Line 43-47)

Although research has shed light on IBS pathophysiology, therapeutic interventions remain symptom driven, employing both pharmacological and nonpharmacological approaches [7]. Several treatment options exist for individuals with IBS, including medication, exercise, fiber supplements, stress management, and psychotherapy [14]. The American College of Gastroenterology (ACG) performed a systematic review to determine the efficacy of 11 IBS therapies, both pharmacological and nonpharmacological, compared with placebo or no treatment. In the highlights of ACGS¸ there is evidence to support the use of antidepressants and psychological therapies for IBS [8]. The British Society of Gastroenterology guidelines[12]on the IBS mechanism and practical management concluded that more accurate means of identifying IBS-C/ IBS-D are required in order to provide the most appropriate treatment. Cochrane review of the efficacy of psychological interventions found that cognitive behavioral therapy (CBT) and interpersonal psychotherapy may benefit patients with IBS, although some issues such as sample size, and clinical heterogeneity needed to be improved [15]. Another systematic review and meta-analysis, which included 30 studies on the effect of psychological therapies on patients with IBS, also found some beneficial effects of CBT and other multi-component psychological therapy [16]. (Line 59-73)

(2)   Moreover, the American College of Gastroenterology (2014 ) performed a

systematic review to determine the efficacy of pharmacologic and nonpharmacologic IBS therapies. Only two therapies overall received strong recommendations for use and were supported, respectively, by evidence of high and moderate quality: linaclotide and lubiprostone for the treatment of IBS-C. 

  Reply (5)Thanks for the thoughtful comments. We have added more detailed information to address the issues of intervention for IBS.   

Although research has shed light on IBS pathophysiology, therapeutic interventions remain symptom driven, employing both pharmacological and nonpharmacological approaches [7]. Several treatment options exist for individuals with IBS, including medication, exercise, fiber supplements, stress management, and psychotherapy [14]. The American College of Gastroenterology (ACG) performed a systematic review to determine the efficacy of 11 IBS therapies, both pharmacological and nonpharmacological, compared with placebo or no treatment. In the highlights of ACGS¸ there is evidence to support the use of antidepressants and psychological therapies for IBS [8]. The British Society of Gastroenterology guidelines on the IBS mechanism and practical management concluded that more accurate means of identifying IBS-C/ IBS-D are required in order to provide the most appropriate treatment[12]. Cochrane review of the efficacy of psychological interventions found that cognitive behavioral therapy (CBT) and interpersonal psychotherapy may benefit patients with IBS, although some issues such as sample size, and clinical heterogeneity needed to be improved [15]. Another systematic review and meta-analysis, which included 30 studies on the effect of psychological therapies on patients with IBS, also found some beneficial effects of CBT and other multi-component psychological therapy [16]. Another study hypothesized the efficacy of CBT for treatment of IBS but found no difference, between CBT, relaxation or standard clinical care [17]. These findings suggest that further investigations are required to develop a richer understanding of the effectiveness of psychological therapy for IBS patients to better control their health outcomes. (Line 59-76)

(3)   Page 2 line 51/52  relaxation training and CBT are not among the most efficient

Methods.

Reply (6) Thanks for pointing it out. The sentence was deleted.  

(4)   secondly the authors probably mean the most effective and not the most efficient 

Reply (7) We have replied in remark #(6)

(8)I think it's important that the tone of the introduction is changed; at the moment it

is very women unfriendly; the authors are blaming women for their IBS; also the

authors seem to confuse correlation and causation. 

Reply (8) In the present study, we purposely chose female nursing students as research participants based on three reasons. First, statistics indicated that women have higher IBS prevalence than men. Second, IBS affect people of all ages, but it’s more likely for people in their teens through 40s. Third, studies that examined the effects of ICBT on IBS for female students are limited. We hope that the study results could be used to inform the development and delivery of interventions targeting female adolescents with IBS problems. In the revised manuscript, we have added

(9)The authors mention the review by the Laird; this review advised to use active

control conditions to control for non-specific treatment effects. Unfortunately this study didn't do that but used a waitlist control group. 

Reply (9) In our study, we made the comparison between three groups, they ICBT, wait list control, and expressive writing. The expressive writing served as an active control since research indicated that expressive is useful in reducing anxiety for emotionally expressive IBS patients. We also mentioned in the Introduction that the effects of expressive writing to decrease anxiety and related emotional reactions has been confirmed. In 2.4 intervention, we replace EW by EW active control group.(Line 195)

(10)The authors fail to mention the problems of a waitlist control group. this is of

particular importance because knowledge that one is not receiving treatment, affects outcomes negatively. Assignment to no treatment may strengthen participants' beliefs that they will not improve, thereby reducing the chance of spontaneous improvement. Participants randomized to waitlist control conditions may improve less than would be expected compared to participants not enrolled in a trial. Using waitlist or no treatment control conditions can lead to the overestimation of the effectiveness of a treatment (Mohr et al., 2009).

A Cochrane review of non-pharmacological interventions for functional

syndromes (14 of the 21 RCTs used CBT) (Van Dessel et al., 2014), and a recent

meta-analysis of 33 RCTs of mindfulness-based interventions (Dunning et al.,

2018), both not only concluded that most benefits disappear when an active

control group is used, instead of a waitlist or no treatment control group. But also,

that there's evidence too that the remaining effects are inflated by bias and

 multiple methodological concerns, including high drop-out rates and selective

biases in sampling.

Reply (10) Thanks for the thoughtful comments. Indeed, if participants randomized to waitlist control conditions might improve less than would be expected compared to participants enrolled in a trial. If use waitlist or no treatment control only, the treatment effect could be overestimated. In our study, in addition to the wait-list control, we also had one active control group, expressive writing. The effectiveness of expressive writing on IBS or symptoms of emotion were investigated in other empirical studies. 

(11)Page 3 line 100 the study uses a Cluster randomised controlled trial and there are

some pitfalls of this sort of trial; for example a reduced statistical efficiency  

relative to randomised controlled trials with the same number of individuals.

what has the study done to address this? 

Reply (11) We have added more detailed information to explain the reason that we chose cluster randomized control trail instead of RCT. Nursing student are divided into group when they are assigned to practicum sites. If students in the same practicum site but randomized to different comparison groups, they may be influenced with each other, the “contamination” is likely to occur by frequent contact between participants in the same practicum units. We also added the limitation of cluster randomized trails in the text.     

In Taiwan, a five-year nursing college program requires students to take several short-term practicum courses beginning in the third year toward the end of their fifth year. Nursing students are divided into groups when they are assigned to practicum sites. Due to the program structure of the college, a cluster randomized control trial design was employed. The practicum students in the same practicum unit were randomly assigned to one of the three groups. The number of participants in each group was not equal, as the unit of random assignment to a given intervention or control was by practicum group (cluster). (Line 129-135)

The randomized control trials we used in the study, though overcome the possible contamination effects, which is likely to occur by contact among individual participants in different groups, it can be susceptible to some methodological problems, such as the number of participants imbalance between treatment groups. (Line 377-380)

(12)Page 3 line 102 eligible students were randomly assigned to 1 of three groups

but how can it then be that in the two treatment groups there were 48 and 42 participants but in the control group there were 70 ? if the randomisation had been done properly it should have been the same number in each group 

Reply (12) Thanks for pointing this out. The reasons that the number of participants in each group is not equal are addressed in remark # (11). The number of participants in each group was not equal, as the unit of random assignment to a given intervention or control was by practicum group (cluster).

(13)Page four figure 1 at the bottom of the figure please explain what T0, T1, T2 and T3 stand for   

Reply (13) Figure was modified as suggested, and what T0,T1,T2, and T3 stand for were explained.

(14)page five line 125 if the scale was translated by the authors into Chinese this at the same time means that the Chinese version of BSSS is not validated

Reply (14) thanks for point this out. We have added the following translation process

in the first paragraph of Measure.

The three-procedure approach suggested by Brislin (1970) was adopted for translation. The English inventories were first translated by both authors into Chinese, and the Chinese inventory was then back-translated into English by a native English speaker. The authors compared the back-translation version with the original and revised the Chinese inventories as required. Three graduate students were asked to fill out both Chinese and English inventories, and further revision was made when inconsistent responses were found between the two versions of the same inventory. (Line 152-158)

(15) line 155  outcome assessments were held on the 2nd, 6th 18th week 

 what does this mean? where they held during the second week or two weeks

after starting treatment and the same applies to 6 and 18 weeks 

Reply (15) Thanks for pointing this out. The outcome measures were administered at the beginning, and the 2nd, 6th and 18th week of the participation in the study. Text was revised as suggested.  

The participants were required to complete and upload their daily assignments before the deadline (before noon each day). Assessments were held on the 2nd, 6th, and 18th week of after starting participation in the study. (Line 190-192)

(16) page 6 line 185 The Rome III criteria was used as diagnostic criteria

 please change was to were because criteria is plural

Reply (16) Thanks for pointing the grammatical error. Since the diagnostic criteria were addressed in the Methods, this sentence was deleted in the revised text.

(17) as from the beginning of 2016 the Rome IV criteria are being used then please

add when your trial started and patients were recruited because otherwise your trial was outdated before it started.

Reply (17) The study was conducted between 2013 and 2015, and data was collected between July 2014 and July 2015. By then, the Rome IV criteria were not used. We added the time period of data collection in the text.

The data was collected between July 2014 and July 2015. (Line 127)

(18)Page 6 line 192 to 194  the BSSS analysis scores are the following 

24.96 ICBT group, 19.07 EW group and 32.00 control group; the differences are

statistically significant; this basically means that there was a significant

difference in severity between the ICBT group and the control group of 30%

(7.04/24.96) and the difference with the EW group was almost double that.

therefore the participants in the control group had more severe IBS, this means

that the groups were poorly matched 

Reply (18) Indeed. The three groups showed significant differences on IBS at the beginning, which did occur in some cluster randomized control trials or even randomized control trails. The differences were taken into count in our statistical analyses (both GEE and ANCOVA). We added more information to address this issue in the result section.

The effect of baseline was adjusted in the GEE and ANCOVA model for each outcome variables. (Line 246-247)

(19)page 6 197 and 201 differed no significantly, should be differed not significantly 

Reply (19) Thanks for pointing this out. Modified as suggested.

These three mean scores showed no significant difference (p > 0.05) (Line 240, 244)

(20)Page six line 208  it is misleading to say that the ICBT group and EW group

exhibited considerable improvements compared with the control group and then add but the difference was not significant. this sentence should be rewritten into something like there was no statistically significant improvement after ICBT (p=0.341) and EW (p=0.373) compared to the control group at T1 (2 weeks after the start of treatment). please leave the β values away. 

Reply (20) Thanks for pointing it out. Indeed, no significant differences were observed between the three groups, indicating that neither ICBT nor EW was superior to the wait-list control to ease IBS symptoms. We addressed that in the discussion, and the results were revised as suggested.

The BSSS assessment of the ICBT group (adjusted mean=-4.66) and EW group (adjusted mean=-4.69) exhibited some improvements compared to the wait-list control group (adjusted mean=-3.00) at the second week (T1), but was not statistically significant (ICBT: β =-1.66, p=0.341; EW: β = -1.69, p=0.373). At the sixth week (T2), the ICBT group (adjusted mean=-2.24) exhibited lack of improvement compared with the wait-list control group (adjusted mean=-6.72) but the differences were not statistically significant (ICBT: β =4.49, p=0.049; EW: β = -1.04, p=0.673). In the 18th week, the ICBT group (adjusted mean=-9.90) and EW group (adjusted mean=-8.22) exhibited considerable improvements compared with the wait-list control group (adjusted mean=-7.04) as assessed using the BSSS, while remaining statistically insignificant (ICBT: β =-2.87, p=0.136; EW: β = -1.18, p=0.569). (Line 254-263)

(21)the sentences about the improvements after six and 18 weeks should also be changed in a similar way. 

Reply (21) Modified as suggested and replied in remark # 20.

(22)At T2 and at T3 there were also no statistically significant improvements after

ICBT and EW compared to the control group according to the authors themselves. It is not relevant that the ICBT group exhibited a greater change than the control group at 18 weeks because the improvements were not statistically significant. So in other words your study has shown that ICBT and EW are not more effective than the waitlist for IBS symptoms; also known as a null effect. 

Reply (22) Thanks for pointing this out. We corrected that in the results section. We also addressed that in the discussion.

Contrary to our hypothesis, no significant differences were observed between the three groups, indicating that neither ICBT nor EW was superior to the wait-list control to ease IBS symptoms. This finding is similar to the conclusion of a recent meta-analysis [38], in which CBT delivered through the internet, or minimal contact CBT showed no improvement in IBS symptoms.The authors speculated that direct contact between therapists and patients/clients was necessary for psychological therapy to be effective. A number of controlled trial studies demonstrated CBT among other personal contact psychological therapies were beneficial for alleviating symptoms[39,40]. Another explaination is the length of time of the interevention or even the number of psychoanalytic sessions attended by the IBS sufferer. CBT requires consistent guided effort and focus, of which 8-10 weeks is insufficient time to accurately assess the potential for IBS. (Line 320-330)

(23)This also means that the abstract, discussion and conclusion need to be changed accordingly.

Reply (23) We have revised the abstract and the text as suggested.

Abstract 

Results: The ICBT and EW groups exhibited significantly some reduction in anxiety and depression at T2 and T3 compared to the wait-list control group. The EW group exhibited significantly greater reduction in anxiety and depression than the ICBT group at T2. However, the ICBT group demonstrated greater improvements in alleviating anxiety and depression at T3 compared to the EW group. Conclusion: ICBT can effectively reduce anxiety and depression, with no effect on IBS symptoms in young women. (Line 22-27)

Discussion

The cluster randomized controlled trail was conducted to compare the effects of internet delivered cognitive therapy (ICBT) with self-administered expressive writing and wait-list control on IBS and its related emotional symptoms. The results indicated that symptoms of IBS as well as depression and anxiety decreased over time for all three groups with only one exception, the IBS scores increased slightly at the end of the treatment for the ICBT group. We speculated that self-help learning of ICBT may become additional stress while students usually had a great deal of assignments need to be completed shortly after the end of practicum. This may explain why the scores of IBS increased for ICBT group at T2. Contrary to our hypothesis, no significant differences were observed between the three groups, indicating that neither ICBT nor EW was superior to the wait-list control to ease IBS symptoms. This finding is similar to the conclusion of a recent meta-analysis [38], in which CBT delivered through the internet, or minimal contact CBT showed no improvement in IBS symptoms.The authors speculated that direct contact between therapists and patients/clients was necessary for psychological therapy to be effective. A number of controlled trial studies demonstrated CBT among other personal contact psychological therapies were beneficial for alleviating symptoms[39,40]. Another explaination is the length of time of the interevention or even the number of psychoanalytic sessions attended by the IBS sufferer. CBT requires consistent guided effort and focus, of which 8-10 weeks is insufficient time to accurately assess the potential for IBS. (Line 313-330)

Conclusion

ICBT can effectively reduce anxiety and depression in young women while exhibitng no relief for their IBS symptoms. EW while effective in the short term, is eclipsed in effect by ICBT over the longer term. (Line 386-388)

(24)As regards to depression and anxiety, it is no surprise that CBT is more effective

than leaving people on the waitlist as a meta analysis by Tolin showed that CBT

is the most effective treatment for both problems.

Reply (24) Indeed. The treatment effects of ICBT on depression and anxiety are well established. In the discussion we added more information to explain the possible mechanism of ICBT on alleviating depression and anxiety.   

ICBT can effectively treat anxiety and depression but not IBS symptoms. The results of this study are similar to the Boyce study [17] which supports that the proposed cognitive model appears to have its effect by altering the cognitive response to visceral hypersensitivity. The Cochrane review of non-pharmacological interventions for functional syndromes (14 of the 21 RCTs used CBT) showed excellent effect when CBT was learned by the participants [41]. It is known, that at least half of IBS patients suffer from anxiety, depression or some form of hypochondriacal manifestation of mental illness. Studies from terriary care suggested that more than 60% of IBS sufferers were in fact suffering from other psychiatric disorders- most commonly anxiety or depressive disorder [42,43,44].The cognitive mechanism of effect between CBT and the emotional symptoms of anxiety and depression are currently being researched. Understanding the underpinnings of the possibility that the mind affects the physical experience and health within an individua;l suggests strongly, that CBT, when it is also similarly refined in its practice, may prove to be a powerful tool in the health care arsenal for both physican and patient. (Line 331-343)

Greater differences between ICBT and the other two methods were observable particularly at the 18th week, where ICBT exhibited notable efficacy. The results of this study are in agreement with the results of previous studies [45, 46, 47, 48]. ICBT appears to exert its effects on anxiety and depression by enhancing the top-down cognitive control of emotions (e.g. prefrontal regions) and/or decreasing the bottom flow of fear-related emotions from limbic structures. Cognitive-behavioral control appears to lead to increased awareness, improved emotional health and con-comittant physical health over the long term. (Line 344-350)

(25)Table 1 please explain what intercept, group, time, group time and baseline mean

 what is DF  what is bald x2. I think everybody understands that p<0.001 means

p<0.001 so what's the point in adding asterixis after the p values? 

Reply (25) We have added detailed information to explain the terms being used in the table.

BSSS=Bowel Symptom Severity Scale (BSSS), STAI-S= The State-Trait Anxiety Inventory, CES-D=The Center for Epidemiological Studies Depression Scale, Intercept= intercept term, Group= treatment group, time= time points, Group*time= interaction term between treatment group and time points, baseline= BSSS, STAI-S and CES-D measures at baseline (T0), wald x2= test of hypotheses on parameters estimated by maximum likelihood, df= degree of freedom. (Line 294-298)

(26) table 2 why are there no P values for control?  what does baseline mean?

 Under T2 why are there no beta entries at baseline? what does T1, weeks 2 mean

do you mean during the second week? please clearly state what it means

 same applies to T2 and T3 

Reply (26) Thanks for pointing this out. We have added detailed information to explain the terms being used in the table. The values of β, SE and P for the control are 0. We can examine the differences between the treatment trails and control by looking at theβ, SE and P values of ICBT and EW.  

baseline= T0, T0= baseline assessment, T1= assessment at 2 weeks, T2= assessment at 6 weeks and end of practicum, T3= assessment at 18 weeks, *p< 0.05, **p<0.01< span=""> (Line 301-302)

(27)Figure 2 is misleading as the BSSS changes at T1, T2 and T3 are not statistically

significant compared to the control group.

Reply (27) Figure 2 showed the trend of the BSSS change at T1, T2, and T3. The scores did decrease over time for all three groups with only one exception, the IBS scores increased slightly at the end of the treatment for the ICBT group. In the results and discussion, we indicated that although the score declined over time, they were not statistically significant.

(28)Page 10 discussion.  I thought this was a study about IBS yet suddenly in the first

 line of discussion it is called gastrointestinal syndrome and not IBS ? please be

consistent and change it to IBS 

Reply (28) Thanks pointing this out. The whole paragraph of discussion was rewritten, the sentence was deleted.      

(29)there are no statistically significant differences between ICBT, EW and the

control group at T1, T2 and T3 therefore this study has shown that neither ICBT nor EW are an effective treatment for IBS.

Reply (29) Indeed. We have replied this in remark # 24

(30)Line 276  symptoms were not resolved; they had improved in all three groups

with no statistically significant differences between the three groups 

Reply (30) Indeed. The symptoms of IBS were decreased for all the groups but not significantly different between the three groups. We have revised as suggested and replied in remark # 24.  

(31)Line 277 it is not difficult to observe the efficacy of ICBT in the short term for the simple reason that there was no statistically significant difference in improvement between the three groups at any of the three points 

Reply (31) Indeed. There was no statistically significant difference between the three groups on IBS, however , ICBT and expressive writing active control did demonstrate effects on depression and anxiety at different time point. We indicated that in the results and provided interpretation in the discussion. The statement was rephrased as below.  

Although the effect of ICBT was not apparent for anxiety and depression in the short term, it superseded EW By the end of the intervention interval. (Line 359-360)

Study limitations 

(32)there are disadvantages of a cluster randomised controlled trial which should be mentioned here too.

Reply (32) Thanks for pointing this out. We have replied in remark # (11).

(33)it was a nonblinded study using subjective outcomes; this combination is

unreliable and likely to lead to false positive results. 

Reply (33) Indeed. Thanks for pointing that out. We have addressed this issue in the limitation.

Inadequate participant blinding is another limitation which may result in placebo or nocebo effects on treatment outcomes. ( Line 381-382)

(34)the study used a Chinese version of BSSS yet this was not validated

Reply (34) thanks for point this out. We have added the following translation process

in the first paragraph of Measure.

The three-procedure approach suggested by Brislin (1970) was adopted for translation. The English inventories were first translated by both authors into Chinese, and the Chinese inventory was then back-translated into English by a native English speaker. The authors compared the back-translation version with the original and revised the Chinese inventories as required. Three graduate students were asked to fill out both Chinese and English inventories, and further revision was made when inconsistent responses were found between the two versions of the same inventory. (Line 152-158)

(35)it is odd that everybody in the three groups who started the trial also finished it.

Were patients super motivated to receive ICBT or were they offered an incentive if they fulfilled all three assessments ?

Reply (35) It is our observation during the research that nursing students are very much open-minded about new experiences and eager to learn skills to cope with stress. We added some information to address no extra credit or incentive was awarded in 2.1. Participants and Eligibility criteria.

They were also informed that the participant was on a voluntary basis and no incentive or extra credit was awarded. (Line 124-125)

(36)a major limitation of the study was not only the use of a waitlist control group

but also the fact that patients in the control group had more severe IBS then in

the other two groups. This means that the groups were poorly matched. this is

also shown by the unevenly distribution of participants (48, 42 and 70). this

suggests that there was a major problem with randomisation  

Reply (36) Thanks for careful consideration on (a) waitlist control(no active control) (b) three groups were significantly different at the baseline on IBS (c) number of participants were different for the three groups. The have replied on remark #9, #10, and #18.

(37)The discussion needs to be rewritten and the conclusion should be that ICBT and EW might reduce symptoms of depression and anxiety but are not effective treatments for IBS 

Reply (37) The abstract and discussion had been revised following the precious comments. We have added three new paragraphs to address the issues of no treatment effects of ICBT on IBS, and the possible mechanism of ICBT on depression and anxiety.

(38)page 11 line 288 to 290 what does this sentence mean? 

Reply (40) We have revised the whole section of the limitation.

  Conclusion 

(39)as mentioned before the conclusion needs to be rewritten and the same applies to

the abstract

Reply (39) Thanks for the precious comments. We have rewritten the abstract and sections of results, discussion, and conclusion. We also replied in remark #22, #23, and #24.
